# Consistency of satellite-based precipitation products in space and over time compared with gauge observations and snow-hydrological modelling in the Lake Titicaca region

Frédéric Satgé[1], Denis Ruelland[2], Marie-Paule Bonnet[3], Jorge Molina[4], Ramiro Pillco[4]

[1]CNES, UMR Hydrosciences, University of Montpellier, Place E. Bataillon, 34395 Montpellier Cedex 5, France
[2]CNRS, UMR Hydrosciences, University of Montpellier, Place E. Bataillon, 34395 Montpellier Cedex 5, France
[3]IRD, UMR Espace-Dev, Maison de la télédétection, 500 rue JF Breton, 34093 Montpellier Cedex 5, France
[4]Universidad Mayor de San Andres, calle 30 Cota Cota, La Paz, Bolivia

*Correspondence to*: Frédéric Satgé (frederic.satge@gmail.com), Denis Ruelland (denis.ruelland@um2.fr)

**Abstract.** This paper proposes a protocol to assess the space-time consistency of 12 satellite-based precipitation products (SPPs) according to various indicators including: (i) direct comparison of SPPs with 72 precipitation gauges; (ii) sensitivity of streamflow modelling to SPPs at the outlet of four basins; and (iii) the sensitivity of distributed snow models to SPPs using a MODIS snow product as reference in an unmonitored mountainous area. The protocol was applied successively to four different time windows (2000–2004, 2004–2008, 2008–2012 and 2000–2012) to account for the space-time variability of the SPPs and to a large dataset composed of 12 SPPs (CMORPH–RAW v.1, CMORPH–CRT v.1, CMORPH–BLD v.1, CHIRP v.2, CHIRPS v.2, GSMaP v.6, MSWEP v.2.1, PERSIANN, PERSIANN–CDR, TMPA–RT v.7, TMPA-Adj v.7 and SM2Rain–CCI v.2), an unprecedented comparison. The aim of using different space-time scales and indicators was to evaluate whether the efficiency of SPPs varies with the method of assessment, time window and location. Results revealed very high discrepancies between SPPs. Compared to precipitation gauge observations, some SPPs (CMORPH–RAW v.1, CMORPH–CRT v.1, GSMaP v.6, PERSIANN, TMPA–RT v.7) are unable to estimate regional precipitation whereas the others (CHIRP v.2, CHIRPS v.2, CMORPH–BLD v.1, MSWEP v.2.1, PERSIANN–CDR and TMPA–Adj v.7) produce a realistic representation despite recurrent spatial limitation over regions with contrasted emissivity, temperature and orography. In nine out of ten of the cases studied, streamflow was more realistically simulated when SPPs were used as forcing precipitation data rather than precipitation derived from the available precipitation gauge networks whereas the SPP's ability to reproduce the duration of MODIS-based snow cover resulted in poorer simulations than simulation using available precipitation gauges. Interestingly, the potential of the SPPs varied significantly when they are used to reproduce gauge precipitation estimates, streamflow observations or snow cover duration and depending on the time window considered. SPPs thus produce space-time errors that cannot be assessed when a single indicator and/or time windows is used, underlining the importance of carefully considering their space-time consistency before using them for hydro-climatic studies. Among all the SPPs assessed, MSWEP v.2.1 showed the highest space-time accuracy and consistency in reproducing gauge precipitation estimates, streamflow and snow cover duration.

# 1. Introduction

## 1.1 On the need for and difficulty involved in estimating precipitation fields

Water resources are facing unprecedented pressure due to the combined effects of population growth and climate change. In the 20th century, water extraction underwent a six-fold increase to sustain food needs and economic levels due to the increasing world population (vision, water council, 2000). At the same time, global warming has led to the redistribution of precipitation, which has favoured the occurrence of both drought and extreme flood events (Trenberth, 2011).

As a key component of the hydrologic cycle, it is therefore crucial to have accurate precipitation estimates in many research fields including hydrological and snow modelling (e.g. Hublart et al., 2016), climate studies (e.g. Espinoza Villar et al., 2009), extreme flooding (e.g.  Ovando et al., 2016) drought (e.g. Satgé et al., 2017a), and monitoring to understand past and ongoing changes and to optimise water resources management (e.g. Fabre et al., 2016, 2015).

Measurements of precipitation are usually retrieved from point gauge stations. Considered as ground truth at the point level, precipitation estimates are then spatialized to represent the distribution of precipitation in space and over time to be used as inputs for impact modelling. However, in most cases, the gauge network is too sparse and unevenly distributed to correctly capture the spatial variability of precipitation. This is especially true for remote regions such as tropical forests, mountainous areas and deserts where the usual insufficient installation and maintenance operations seriously compromise precipitation monitoring. An alternative approach consists in using precipitation derived from weather radar using the backscattering of electromagnetic waves via hydrometeors (e.g. Mahmoud et al., 2018). Unlike measurements using traditional gauges, this technique monitors large areas in a distributed way thus offering the opportunity to monitor precipitation over remote regions. However, ground radar measurements are rarely available at the global or even regional scale (Tang et al., 2016) and are limited in the case of complex terrain which interferes with the radar signal (Zeng et al., 2018). More recently, some authors (Messer et al., 2006; Overeem et al., 2011; Zinevich et al., 2008) reported on the possibility to estimate precipitation from wireless networks such as commercial cellular phone microwave links. These estimations are based on the attenuation of the electromagnetic signals between telecommunication antennas during precipitation events and their first results are promising (see e.g. Doumounia et al., 2014). However, this technique faces the problem of private cellular phone company policy about sharing data and telecommunication antenna are mainly located in urban areas, which limits accurate precipitation estimates in space. On the other hand, several satellite-based precipitation estimates (SPPs) are now available, making possible to monitor precipitation on regular grids at the near global scale, representing an unprecedented opportunity to complement traditional precipitation measurements.

## 1.2 Satellite-based precipitation estimates (SPPs): opportunities and limitations

Several SPPs have become available in recent decades to monitor precipitation at global scale and on regular grids. The first generation of SPPs appeared with the Tropical Rainfall Measuring Mission (TRMM) launched in 1997 by NASA (National

Aeronautics and Space Administration) and the Japan Aerospace Exploration Agency (JAXA). Over the last 18 years, the TRMM Multisatellite Precipitation Analysis (TMPA) (Huffman and Bolvin, 2014), the Climate prediction centre MORPHing (CMORPH) (Joyce et al., 2004), the Precipitation Estimation from remotely Sensed Information using Artificial Neural Networks (PERSIANN) (Sorooshian et al., 2000) and the Global Satellite Mapping Precipitation (GSMaP) (GSMaP, 2012) SPP datasets have been developed based on the TRMM mission to deliver precipitation estimates at the 0.25° grid scale. In 2014, the Global Precipitation Measurement (GPM) mission was launched to ensure TRMM continuity. The second generation of SPPs based on GPM missions included the Integrated Multi-SatellitE Retrievals for GPM (IMERG) (Huffman et al., 2017) and a new GSMaP version product which deliver precipitation estimates at a finer grid scale (0.1°) than the first generation of SPPs but estimates are limited to the period from 2014 to the present. At the same time, some SPPs took advantage of previous SPPs and missions to estimate precipitation over larger time window: long-term SPP generation. This is the case of PERSIANN-Climate Data Record (PERSIANN-CDR) (Ashouri et al., 2015), Multi-Source Weighted-Ensemble Precipitation (MSWEP) (Beck et al., 2017) and Climate Hazards Group InfraRed Precipitation (CHIRP) with Station data (CHIRPS) (Funk et al., 2015).

However, SPPs are indirect measurements made from satellite/sensor constellations, including passive microwaves (PMW) and infra-red (IR) sensors on board low earth orbital (LEO) and geosynchronous satellites and are subject to uncertainty due to technical limitations. Indeed, the irregular sampling and limited overpass of LEO PMW measurements impede the correct capture of short term and slight precipitation events (Gebregiorgis and Hossain, 2013; Tian et al., 2009) which can introduce error in precipitation estimates over arid regions and/or during the dry seasons (Prakash et al., 2014; Satgé et al., 2017a; Satgé et al., 2015; Shen et al., 2010). In mountainous regions, the precipitation /no precipitation cloud classification based on cloud top IR temperature may fail in the case of precipitation processes resulting from orographic warm clouds (Dinku et al., 2010; Gebregiorgis and Hossain, 2013; Hirpa et al., 2010). The contrast between temperature and emissivity (i.e. water and snow covered area) of rough land surfaces creates background signals similar to those produce by precipitation, leading to misinterpretation between rainy or not rainy clouds, which can introduce high bias in precipitation estimates (Satgé et al., 2016; Tian and Peters-Lidard, 2007; Ferraro et al. 1998; Hussain et al. 2016; Satgé et al., 2018).

In addition to these spatial inconsistencies, the orbital satellite context implies constantly varying input data for each observation time (snapshot), which likely introduces inhomogeneity in the SPP time records. This could be exacerbated with aging sensors and permanent sensor failures. As an example, the TRMM satellite mission ended on April 8, 2015, making unavailable the TRMM Microwave Imager (TMI) from input data used for TMPA retrieval (Huffman and Bolvin, 2016). Consequently, the potential of SPPs is expected to present space and time errors whose quantification is crucial before their use for hydro-climatic studies.

## 1.3 State of the art evaluation of SPPs

In the context described above, many studies have reported on the efficiency of SPPs over different regions. The most common way to evaluate SPP potential is to compare their estimates with precipitation gauge measurements, as reviewed by Maggioni et al. (2016) and Sun et al. (2017). The comparison of gauge-based assessment studies confirmed the spatial variability of SPP efficiency in reproducing precipitation, so no single SPP can be said to be the most effective one at global scale. For example, when TMPA, CMORPH, and PERSIANN SPP datasets were compared, TMPA was found to be closer to the observed precipitation in India (Prakash et al., 2014), the Guyana shield (Ringard et al., 2015), Africa (Serrat-Capdevila et al., 2016), Chile (Zambrano-Bigiarini et al., 2017) and South America Andean plateau (Satgé et al., 2016), whereas CMORPH was closer to observed precipitation in Bali, Indonesia (Rahmawati and Lubczynski, 2017), Pakistan (Hussain et al., 2017), and China (Su et al., 2017; Zeng et al., 2018). However, these assessments based on comparison with gauge observations did not assess SPP's potential performance over unmonitored regions. This is especially true for high mountainous regions where available gauge networks (generally located in the valley) cannot correctly represent the local precipitation induced by topographic effects. As a result, evaluating SPP potential over high mountainous regions remains challenging (Hussain et al., 2017; Satgé et al., 2017b).

An alternative method consists in assessing the sensitivity of hydrological models to SPPs. The efficiency of SPPs can be evaluated indirectly via their ability to generate reasonable discharge simulations at the outlet of the basin concerned. Compared to gauge-based assessment studies, fewer authors have reported on hydrological sensitivity to SPPs, as reviewed in (Maggioni and Massari, 2018). For example, TMPA, CMORPH and PERSIANN datasets were compared as forcing data for hydrological modelling in Africa (Casse et al., 2015; Thiemig et al., 2013; Tramblay et al., 2016) and South America (Zubieta et al., 2015). These studies provided complementary information to gauge-based assessments, offering an operational overview of SPPs for the management of water resources. However, due to the aggregation process at basin scale, the potential of SPPs over specific ungauged regions remains unclear. Moreover, in these studies, the SPPs were not compared with gauge observations (Thiemig et al., 2013) or provided only a brief comparison at basin (Casse et al., 2015; Tramblay et al., 2016) or gauge (Zubieta et al., 2015) scale. Therefore it is difficult to conclude on the respective advantages and limitations of using gauges or streamflow data as indicators to assess the ability of SPPs to reproduce precipitation patterns as SPPs could rank differently depending on the indicator used. For example, considering CMORPH, PERSIANN and TMPA datasets over two African watersheds, TMPA showed the closest estimate in comparison with gauges for both basins (Thiemig et al., 2012) while CMORPH and TMPA provided more accurate streamflow simulations depending on the basin considered (Thiemig et al., 2013).

Whatever the selected approach (based on gauges and/or hydrological modelling), the analysis is performed using a single time window which does not assess the temporal variability of SPPs due to the acquisition process and/or aging sensors. To date, only a few studies have been conducted to observe changes in the efficiency of SPPs over time. For example, CHIRPS precipitation estimates were analysed over two distinct time windows to assess potential changes in precipitation accuracy over Cyprus and Nepal from one period to another (Katsanos et al., 2016; Shrestha et al., 2017). Similarly, Bai et al., 2018 analysed the accuracy of CHIRPS in mainland China separately for each year to assess its inter-annual variability. These

studies highlighted temporal inconsistencies in CHIRPS estimates inherent to variations in the input data used for precipitation retrieval. However, only the CHIRPS SPP was considered and similar features are to be expected with other SPPs.

Today more than 20 SPPs are available from the first (TRMM), second (GPM) and long-term SPP generation (Beck et al., 2017). Nevertheless previous studies only considered SPP subsets. SPP assessment has indeed focussed on: (i) a single or a limited sample of SPPs (e.g. Cao et al., 2018; Erazo et al., 2018; Shrestha et al., 2017); (ii) transition from the previous to a new version of an algorithm for a specified SPP (TMPA-v6 to TMPA-v7 for example) (e.g. Chen et al., 2013; Melo et al., 2015; Milewski et al., 2015); and (iii) the effectiveness of the transition from the first (TRMM) to the second (GPM) generation of SPPs (e.g. Satgé et al., 2017; Sharifi et al., 2016; Wang et al., 2017). All these studies provided useful feedback related to their specific objectives but did not really help assess the respective performance of SPPs due to the small sample of SPPs considered.

For these reasons, comprehensive feedback on SPPs including space-time consistency, different indicators, insights over unmonitored regions, and a representative SPPs sample can only be acquired by backcrossing large SPP assessment studies. Even so, as each study is based on different statistical indices, spatial and temporal scales and periods, such an effort is seriously compromised.

## 1.4 Objectives

From the previously established state of the art, this paper investigates the influence of selected indicators and time windows on assessments of the space-time consistency of SPPs. The comprehensive protocol relies on different indicators: gauge observations; (ii) observations of streamflow using sensitivity analysis of a lumped hydrological model in different catchments; and (iii) snow cover observed from satellite imagery via sensitivity analysis of a distributed snow model in an unmonitored mountainous area, applied to four time windows. The aim of using different indicators was to evaluate whether the efficiency of the SPPs varies with the assessment method, whereas different time windows are used to evaluate a potential variation in SPP performance over time. The Lake Titicaca region was selected as study area because it includes all the specific features considered as potential limiting factors for SPPs (high mountain massifs, large water bodies and snow covered areas) to evaluate the potential of SPPs in an extreme context in terms of the sensors' limitations with respect to the orographic effect (i.e. mountains) and high temperature/ emissivity contrast (i.e. Lake Titicaca and a snow-covered region). It also offers the opportunity to provide feedback on the use of SPPs over poorly monitored regions

## 2. Material: study area and data

### 2.1. Study area

The Lake Titicaca basin is located between 14°S and 17°S and 71°W and 68°W in the northern part of the South American Andean plateau, known as the Altiplano. It extends over an area of 49,000 km$^2$. The Lake Titicaca catchment is bordered to the west and east by the two Cordilleras (Occidental and Real) and includes a few snow-covered areas. With a surface area of

8,560 km$^2$, a mean depth of 105 m (284 m max) and a water volume estimated at 903 km$^3$ (Delclaux et al., 2007) Lake Titicaca is the main water body and source of the endorheic Altiplano hydrologic system. The Lake Titicaca is drained by the Desaguadero River to the south (Fig. 1) which contributes up to 65% of water inflows into Lake Poopó (second largest Bolivian lake) (Pillco and Bengtsson, 2010). An accurate Lake Titicaca water balance for monitoring purposes is therefore crucial to support efficient water resources management in the Altiplano. However, the transboundary, economic and remote context means hydro-meteorological monitoring is sparse. Thanks to almost global scale coverage, SPPs represent a promising alternative to monitor regional precipitation in space and over time, and offer unprecedented opportunity to achieve efficient regional water resources management.

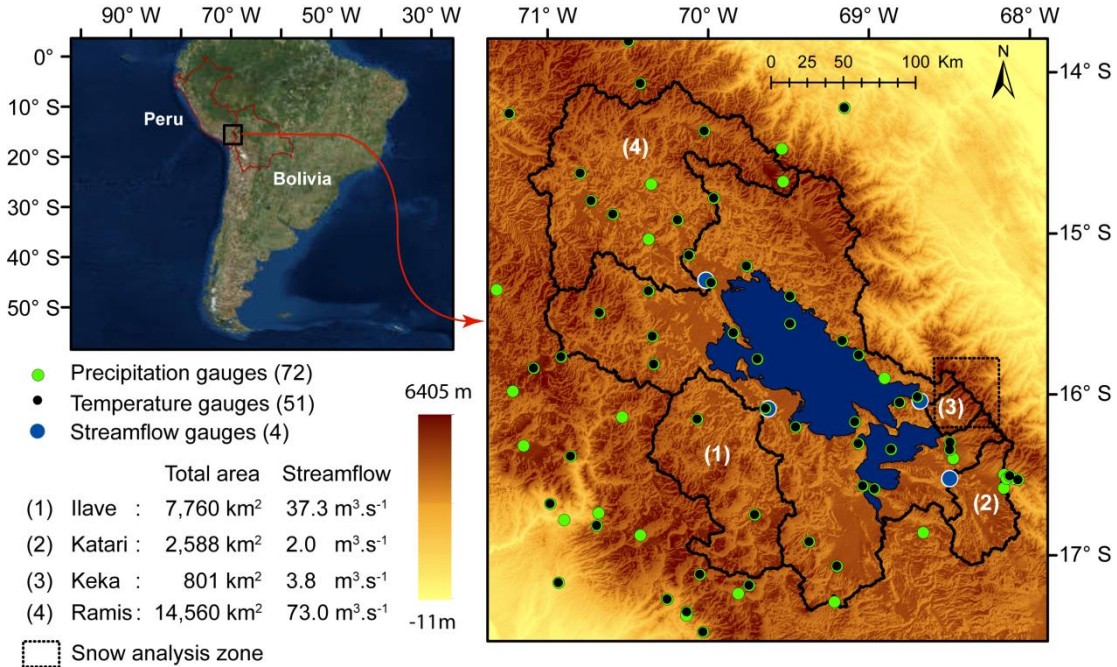

**Fig. 1 Study area: location, meteorological stations (precipitation and temperature observations), streamflow gauges, studied catchments and snow analysis zone.**

### 2.2. Hydro-climatic data

### 2.2.1. Hydro-meteorological stations

Precipitation and air temperature data for Bolivia were provided directly by the *Servicio Nacional de Hidrologia e Meteorologia* (SENAMHI) whereas for Peru, data were collected from the Peruvian SENAMHI website. Only weather gauges with less than 20% daily missing data over the 2000–2012 period were selected, giving a total of 72 stations in Bolivia and 51 in Peru (Fig. 1).

Water level and discharge data from SENAMHI were managed with the free software HYDRACCESS (Vauchel, 2005) developed by SO-HYBAM to obtain daily discharge records at the outlet of four catchments (Fig. 1): the Ilave (7,766 km²),

the Katari (2,588 km²), the Keka (801 km²), and the Ramis (14,560 km²) catchments with respective mean annual discharge estimated at 37.3, 2.0, 3.8 and 73.0 m$^3$.s$^{-1}$ (Uria & Molina, 2013). Nearly complete discharge observations over the 2000–2012 period were available for two basins (Katari and Keka), whereas discharge observations were only available from 2008 to 2012 for the other two (Ilave and Ramis).

### 2.2.2. Interpolation of meteorological in-situ observations

To obtain continuous and spatialized meteorological series for the study area, precipitation and temperature gauge data were interpolated using the inverse distance weighted method (IDW) on a 5-km grid at the regional scale (for details on the purpose of hydrological modelling, see section 3.2) and on a 500-m grid at the local scale (for the purpose of snow modelling, see section 3.3). The choice of the IDW technique as interpolation method was based on the study of Ruelland et al. (2008), which showed low sensitivity of  hydrological models to rainfall input datasets derived using different interpolation methods (IDW, Thiessen, spline, ordinary kriging) with IDW yielding the highest hydrological efficiency. Temperature values were interpolated by accounting for a constant lapse rate of 6.5°C km$^{-1}$, in a similar way to that described in Ruelland et al. (2014). Because the gauges are mainly located in the flat land part of the basins, it was not possible to provide evidence for an effect of elevation on precipitation distribution. Consequently, no orographic effect was accounted for in the interpolation of the point precipitation observations. $P_{ref}$ and $T_{ref,}$ refer to interpolated precipitation and temperature, respectively.

### 2.3. Remote sensing data

### 2.3.1. Satellite precipitation estimates (SPPs)

Twelve SPPs with a spatial resolution below or equal to 0.25° (~25 km at the Equator) were selected for the 2000–2012 period. Other precipitation datasets with coarser resolution (>0.25°) are currently available but we did not use them because: (1) the scarce available gauges network will not warrant a consistent potential assessment in reason to the difference between point-gauge and grid-cell-average measurement (Tang et al., 2017) and (2) the considered catchments and snow analyses zone area is smaller than such coarse resolution precipitation datasets.  However, it is worth mentioning that in specific situation, coarse resolution SPPs could perform better than higher resolution SPPs (Beck et al., 2018) and that reanalyses precipitation datasets tend to be a better choice in cold regions/periods (Huffman et al. 1995). Such statements cannot be verified in the present study in reason to the scarce gauge network context and considered catchments and snow analysis zone area.

The SPPs include the following datasets: Climate Hazards Group InfraRed Precipitation (CHIRP), Climate Prediction Center MORPHing (CMORPH), Global Satellite Mapping of Precipitation (GSMaP), Precipitation Estimation from Remotely Sensed Information using Artificial Neural Network (PERSIANN), Soil Moisture to Rain (SM2Rain) method, Tropical Rainfall Measuring Mission (TRMM), Multisatellite Precipitation Analysis (TMPA) and Multi-Source Weighted-Ensemble Precipitation (MSWEP). All the SPPs used a combination of satellite (S) data gathering information from passive microwave (PMW) radiometers and infra-red (IR) data from Low Earth Orbital (LEO) and geosynchronous satellites, respectively except

for the SM2Rain method, which relies on satellite surface soil moisture derived from passive and active microwave. The selected SPPs differ in terms of the combination of satellite sensors, algorithms and whether the products include reanalysis (R) and/or a calibration step against gauge (G) data in their processing or not. Table 1 provides an overview of these SPPs and relevant references for more information on their respective production. The mean annual precipitation pattern retrieved from all SPPs is presented in Fig. 2.

SPPs were first aggregated to obtain daily time step records using 8h to 8h (local time) time windows to match local daily gauge observations. It should be noted that some SPPs (CHIRP v.2, CHIRPS v.2 and GSMaP v.6) are only delivered at daily scale with a daily aggregation based on different time windows which could compromise the comparison of SPPs at daily scale. Finally, using the nearest neighbour technique, all the SPPs were spatially resampled to 5 km to facilitate their comparison. The resulting database consists of 10,500 daily virtual stations at 5 km spatial resolution over the 2000–2012 period for each SPP. Additionally, SPPs were resampled to 500 m resolution over the selected subset region to assess SPP potential for snow modelling (see Fig. 1).

| Full Name | Acronym | Data source | Temporal coverage | Temporal resolution | Spatial coverage | Spatial resolution | References |
|---|---|---|---|---|---|---|---|
| Climate Hazard Group InfraRed Precipitation v.2 | CHIRP v.2 | S,R | 1981–present | daily | 50° | 0.05° | Funk et al., 2015 |
| Climate Hazard Group InfraRed Precipitation with Station v.2 | CHIRPS v.2 | S,R,G | 1981–present | daily | 50° | 0.05° | Funk et al., 2015 |
| CPC MORPHing technique RAW v.1 | CMORPH–RAW v.1 | S | 1998–present | 3 h | 60° | 0.25° | Joyce et al., 2004 |
| CPC MORPHing technique bias corrected v.1 | CMORPH–CRT v.1 | S,G | 1998–present | 3 h | 60° | 0.25° | Joyce et al., 2004 |
| CPC MORPHing technique blended v.1 | CMORPH–BLD v.1 | S,G | 1998–present | 3 h | 60° | 0.25° | Xie et al., 2011 |
| Global Satellite Mapping of Precipitation Reanalyse Gauges v.6 | GSMaP v.6 | S,G | 2000–02/2014 | daily | 60° | 0.1° | Ushio et al., 2009 Yamamoto et al., 2014 |
| Multi-Source Weighted-Ensemble Precipitation v.2.1 | MSWEP v.2.1 | S,R,G | 1979–present | 3h | Global | 0.1° | ==Beck et al., 2017== |
| Precipitation Estimation from Remotely Sensed Information using Artificial Neural Networks | PERSIANN | S,G | 2000–present | 6 h | 60° | 0.25° | Hsu et al., 1997 Sorooshian et al., 2000 |
| PERSIANN-Climate Data Record | PERSIANN–CDR | S | 1983–2016 | 6 h | 60° | 0.25° | Ashouri et al., 2015 |
| TRMM Multi-Satellite Precipitation Analysis Real Time v.7 | TMPA–RT v.7 | S | 2000–present | 3h | 60° | 0.25° | Huffman et al., 2010 Huffman and Bolvin, 2014 |
| TRMM Multi-Satellite Precipitation Analysis Adjusted v.7 | TMPA–Adj v.7 | S,G | 1998–present | 3h | 50° | 0.25° | Huffman et al., 2010 Huffman and Bolvin, 2014 |
| Soil Moisture to Rain from ESA Climate Change Initiative v.2 | SM2Rain–CCI v.2 | S | 1998–2015 | daily | Global | 0.25° | Ciabatta et al., 2017 |

**Table 1: Main characteristics and references of the 12 SPPs considered. In the data source column, S stands for satellite, R for reanalysis, and G for gauge information.**

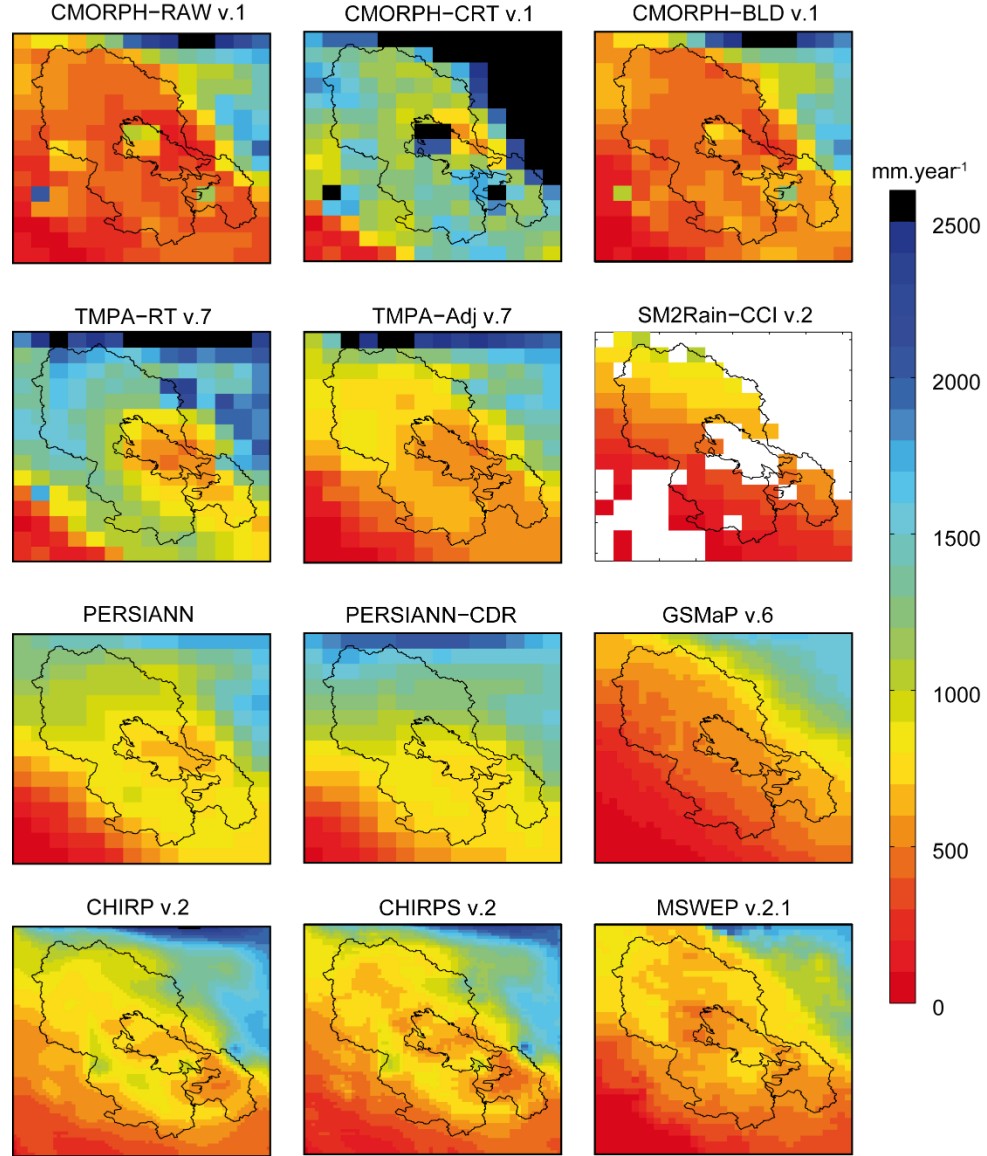

**Figure 2: Mean annual precipitation maps for the 2000–2012 period retrieved from all SPPs at their original grid size. For each SPP, only the grid-cells with more than 80% available daily data were retained. In order to keep the regional precipitation pattern visible, black colour was used to filter grid-cells whose mean annual precipitation was greater than 2,500 mm.year⁻¹**

### 2.3.2. MODIS snow products

MOD10A1 (Terra) and MYD10A1 (Aqua) snow products version 5 were downloaded from the National Snow and Ice Data Center for the period 24 February 2000–6 January 2016. This corresponds to 5,795 daily values among which 5,697 are available for MOD10A1 (98.3%) and 4,918 for MYD10A1 (84.9%) since Aqua was launched in May 2002 and became operational in July 2002. These snow products are derived from a NDSI (Normalized Differential Snow Index) calculated from

the near-infrared and green wavelengths, and for which a threshold has been defined for the detection of snow. Cloud cover represents a significant limit for these products, which are generated from instruments operating in the visible-near-infrared wavelengths.

As a result, the grid-cells were gap-filled so as to produce daily cloud-free snow cover maps of the study area. The different classes in the original products were first merged into three classes: no-snow (no snow or lake), snow (snow or lake ice), no-data (clouds, missing data, no decision, and saturated detector). The missing values were then filled according to a gap-filling algorithm described in Ruelland et al. (submitted). This algorithm works in three sequential steps: (i) Aqua/Terra combination; (ii) temporal deduction by sliding the time filter up to 6 days; (iii) spatial deduction by elevation and neighbourhood filter to gap-fill the remaining no-data grid-cells. The resulting database consists of 8,170 binary (snow/no-snow) daily stations at 500 m spatial resolution for the period 2000–2012 (hereafter $M_{sc}$). Finally, Snow Covered Distribution (SCD) represents the percentage of days with snow and can be retrieved for any grid-cells and periods.

## 3. A protocol to evaluate the space-time consistency of satellite precipitation estimates

Figure 3 is a flowchart of the main methodological steps. Twelve SPPs were first considered as a representative sample of currently available SPPs. This is an important consideration to guide potential SPP users towards the most efficient SPP. However, to avoid overloading the research, a pre-selection was made at the Titicaca Lake catchment scale (hereafter denoted regional scale) to discard less suitable SPPs. The remaining SPPs were then assessed using three successive and complementary methods. The first assessment step consisted of comparing SPPs and gauge observations at the locations of the 69 grid-cells which included gauges. The second assessment step consisted of analysing the sensitivity of streamflow modelling to the SPPs at the four basin outlets using observed streamflow as reference data. The third step consisted of analysing the sensitivity of snow modelling to the SPPs (precipitation datasets) over a subset mountainous area in the Andes (see Fig. 1) using SCD as observed from MODIS gap-filled snow products as reference. Each assessment step was analyzed according to three 4-year time windows (2000–2004, 2004–2008, 2008–2012) and one 12-year time window (2000–2012) in which a hydrological year corresponds to a period starting on 1st of October to the following 30th of September. The aim of the proposed protocol was to investigate the influence of the selected indicator (gauges, streamflow modelling, snow modelling) and time window to assess the SPPs space-time consistency. More details of the proposed protocol are presented in the following sections. It is noteworthy that the use of a 10-day time scale rather than a daily time scale may conceal some of the differences among the datasets, notably by eliminating any insights into their capacity to capture individual events and higher intensities. However, our choice was based on the inconsistencies we expected between gauges and daily measurements of SPPs as a reason to (i) use a different daily time window aggregation than the local one (8 am to 8 pm) for SPPs delivered at daily scale, (ii) the spatial inconsistency between point-gauge measurement and average grid-cell measurement (Tang et al, 2018), and (iii) the temporal filters used for gap filling of MODIS snow products, which led us to consider that these reference data were more valid at a 10-day scale than at a daily scale."

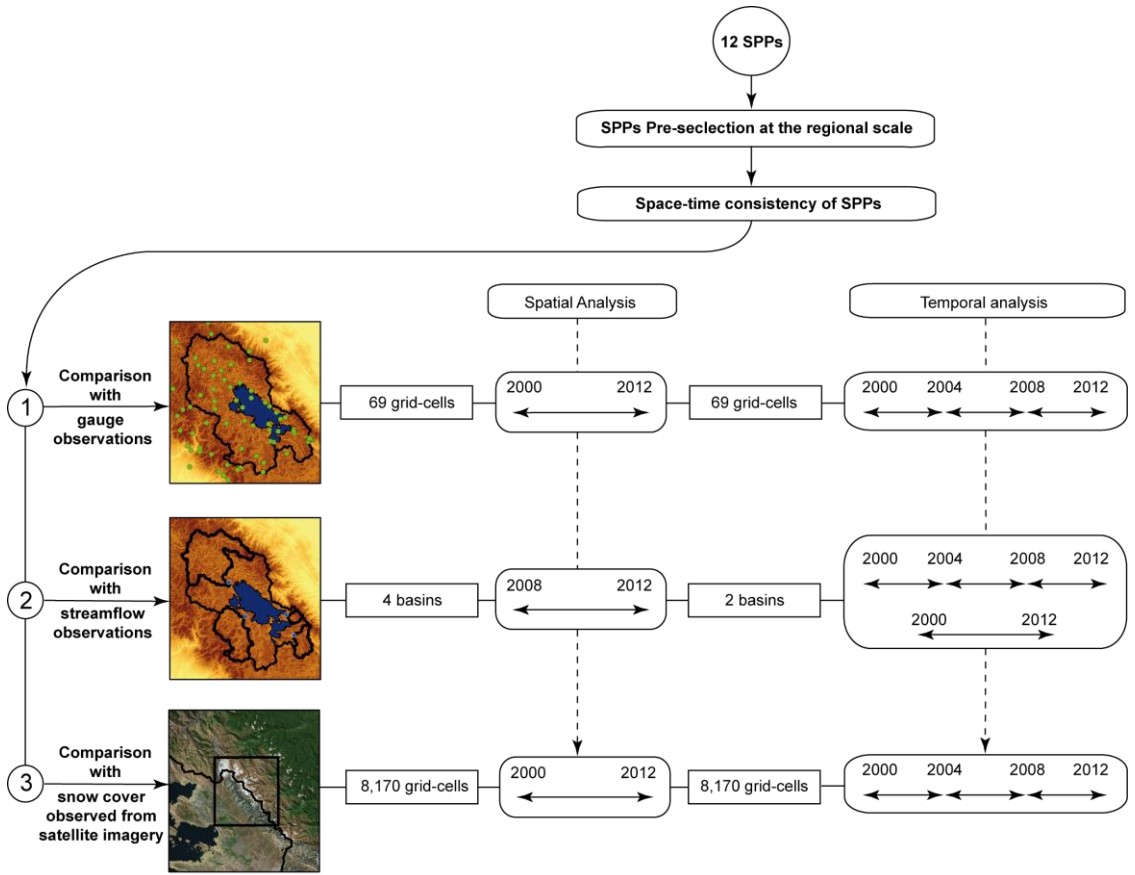

**Figure 3: Flowchart of the main steps from SPP preselection to successive assessment approaches including (1) comparison between SPPs and gauge observations, (2) sensitivity analysis of runoff modelling to the various SPPs at the four basin outlets and (3) sensitivity analysis of snow modelling to the various SPPs over a subset mountainous area in the Andes.**

## 3.1 Comparison of SPPs with gauge observations: pre-selection and evaluation

SPP consistency was first analysed at the regional scale for the 2000–2012 period. For each of the 69 0.05° grid-cells including at least one precipitation gauge, the daily precipitation series from the gauges and SPPs were first aggregated to the 10-day time step over the 12–year period (2000–2012). For each of the 69 0.05° grid-cells, the 10-day records were only computed when more than 80% of daily values were available from all precipitation datasets ($P_{ref}$ and SPPs) for exactly the same date. Next, mean spatially averaged 10-day precipitation series were computed from $P_{ref}$ and all SPPs by aggregating the values from all 69 grid-cells. It should be noted that SM2Rain–CCI v.2 estimates rely on soil moisture observations with many missing data over water bodies and mountainous regions (Dorigo et al., 2015), leading to significant spatial gaps over Lake Titicaca and the Cordillera region (Fig. 3). As a result, in comparison to other SPPs, only 44 $P_{ref}$ 0.05° grid-cells (including gauges)

were available for SM2Rain. Therefore, SM2Rain–CCI v.2 was analysed separately from other SPPs by computing additional $P_{ref}$ and SM2Rain–CCI v.2 mean spatially averaged 10-day precipitation series based on the 44 available $P_{ref}$ 0.05° grid-cells.

Mean spatially averaged 10-day SPPs and $P_{ref}$ series for the 12-year period 2000–2012 were compared according to different statistical criteria, correlation coefficient (CC), standard deviation (STD), percentage bias (%B) and the centred root mean square error (CRMSE) (Eq. 1–4):

$$CC = \frac{Cov(SPP, P_{ref})}{STD_{SPP} \times STD_{ref}} \tag{1}$$

where CC is the correlation coefficient, *SPP* and *$P_{ref}$* are the SPP and $P_{ref}$ precipitation time series, and *Cov* is the covariance.

$$STD = \sqrt{\frac{1}{n}\sum_{i=1}^{n}(P_i - \bar{P})^2} \tag{2}$$

where *STD* is the standard deviation in mm, *n* is the number of values; and *P* the precipitation value in mm (SPP or $P_{ref}$).

$$\%B = \frac{\frac{1}{n}\sum_{i=1}^{n}(SPP_i - P_{ref_i})}{\frac{1}{n}\sum_{i=1}^{n}P_{ref_i}} \times 100 \tag{3}$$

where *%B* is the SPP bias value as a percentage, *n* is the number of values, *SPP* the precipitation estimate of the considered SPP value in mm; and *$P_{ref}$* the reference precipitation value in mm.

$$CRMSE = \sqrt{\frac{1}{n}\sum_{i=1}^{n}((SPP_i - \overline{SPP}) - (P_{ref_i} - \overline{P_{ref}}))^2} \tag{4}$$

where CRMSE is the centred root mean square error in mm, n is the number of values, *SPP* the precipitation estimate of the considered SPP value in mm; and *$P_{ref}$* the reference precipitation value in mm.

To facilitate interpretation of the statistical results, the Taylor diagram (Taylor, 2001) was used to present obtained CC, and normalized values of STD and CRMSE. Normalization was performed by dividing SPPs CRMSE and STD by $P_{ref}$ STD. Therefore, in the Taylor diagram, the reference (black dot) corresponds to CRMSE, STD and CC values of 0, 1 and 1, respectively. Also in the Taylor diagram, the position of the SPPs relative to the reference dot is an integrated indicator (CRMSE, STD, and CC) of SPP efficiency in reproducing gauge precipitation. The shorter the distance between the SPPs and the reference position, the closer the SPPs and $P_{ref}$ estimates. Additionally, %B values were used to observe the potential over/underestimation of each SPP considered. For the following assessment step, only the six SPPs most efficient at the regional scale for the 2000–2012 were considered.

To assess the consistency of SPPs over time, mean spatially averaged precipitation 10-day series were compared to $P_{ref}$ according to a Taylor Diagram for three 4-year periods corresponding to the 2000–2004, 2004–2008 and 2008–2012 periods, respectively. For all the periods considered, 98.6% of $P_{ref}$ grid-cells had more than 90% 10-day records for both the wet and

dry season. Consequently, the temporal assessment of the SPPs was not expected to be influenced by any inconsistency of $P_{ref}$ over time in terms of available records.

To assess the spatial consistency of the SPPs, we compared the CC, CRMSE and %B computed between the SPPs and $P_{ref}$ 10-day series for the 2000–2012 period at the location of each grid-cell which included gauges. For each SPP, CC, CRMSE and %B obtained at the grid-cell level were plotted to highlight regions potentially concerned by low (high) SPP potential. We repeated the analysis for the three 4-year windows. We only considered CRMSE scores to simplify interpretation of the results, as this score was found to provide more statistical discrimination than CC, STD and %B. It is worth mentioning that resampling SPPs (see section 2.3.1) could affect the assessment of SPP potential at the grid-cell level. Indeed, at a coarser spatial resolution (0.25°), SPP grid-cells average precipitation estimates are expected to be more representative of the mean precipitation derived from all the gauges in the grid-cell considered than the one derived from a single gauge. However, for these particular grid-cells, a preliminary SPP assessment at the original and resample grid size revealed no significant differences (data not shown).

Finally, for each grid-cell, mean and STD CRMSE values were computed from the three CRMSE values obtained from the three 4-year periods. Mean and STD CRMSE values were then plotted to assess the consistency of SPPs in space and over time. The lower the mean and STD of CRMSE, the more stable the SPP considered at the specific grid-cell location.

## 3.2. SPPs as input data for hydrological modelling

The GR4j lumped hydrological model (Perrin et al., 2003) was chosen to analyse the sensitivity of streamflow simulations to the various SPPs. The model has demonstrated its ability to perform well under various hydro-climatic conditions (e.g., Coron et al., 2012; Perrin et al., 2003; Grouillet et al., 2016; Dakhlaoui et al., 2017), notably in the Andean region (e.g., Hublart et al., 2016).

This model relies on daily precipitation (P) and potential evapotranspiration (PE) ,which was computed using the formula proposed by Oudin et al. (2005) (Eq. 5):

$$PE = \frac{R_e}{\lambda \rho} \frac{T + 5}{100} \qquad \text{if } (T + 5) > 0 ; \qquad else\ PE = 0 \qquad\qquad (5)$$

where $PE$ is daily potential evapotranspiration (mm), $R_e$ is extra-terrestrial solar radiation (MJ.m$^{-2}$.d$^{-1}$), which depends on the latitude of the target point and the Julian day of the year, $\lambda$ is the net latent heat flux (fixed at 2.45 MJ.kg$^{-1}$), $\rho$ is water density (fixed at 11.6 kg.m$^{-3}$) and T is the daily mean air temperature (°C) estimated at the target point by interpolating the gauge observations while correcting for elevation.

Firstly, a production module computes the amount of water available for runoff, i.e., "effective precipitation". To do so, a soil-moisture accounting (SMA) store is used to separate the incoming precipitation into storage, evapotranspiration and excess precipitation. At each time step, soil drainage is computed as a fraction of the storage and added to excess precipitation to form the effective precipitation. Secondly, a routing function split the effective precipitation into two components: 90% is routed as delayed runoff through an unit hydrograph UH1 in series with a non-linear routing storage while the remaining 10% is routed

as direct runoff through an unit hydrograph UH2 (Perrin et al., 2003). UH1 and UH2 consist in a slow and quick routing path, respectively, to account for differences in runoff delays. Finally, the streamflow at the catchment outlet is computed by summing up delayed and direct runoff. This model relies on four calibrated parameters: maximum capacity of the soil moisture accounting store (X1, mm), inter-catchment exchange coefficient (X2, mm), maximum capacity of routing storage (X3, mm), and time base for unit hydrographs (X4, days) for each catchment. Acceptable parameter bounds were defined according to the recommendation of Perrin et al., (2003) and previous experiments in a similar Andean context (Hublart et al., 2016; Ruelland et al., 2014). The following ranges were used for model calibration with all the precipitation datasets tested ($P_{ref}$ and SPPs): 10 mm < X1 <1,800 mm, -5 mm < X2 < 5 mm, 1 mm < X3 < 500 mm, 0.5 days < X4 <5 days. These ranges are large enough to compensate for the differences in the different precipitation datasets and basins. Beyond these ranges, we assumed that streamflow simulations could not be realistic. In practice, parameter bounds were rarely reached when calibrating the model with the different datasets tested (data not show here for the sake of brevity).

The area catchment P and PE averaged values were computed from the 5-km grid-cells P ($P_{ref}$ and SPPs) and PE included in each catchment considered. We used a weighted average based on the 5 km*5 km fraction included in the catchment considered. $P_{ref}$ and SPPs were used sequentially as forcing precipitation datasets for the streamflow simulation. For each run, model parameter calibration was based on the shuffled complex evolution (SCE) algorithm (Duan et al., 1992) by optimising the Nash-Sutcliffe efficiency criterion (NSE, Eq. 6; Nash and Sutcliffe, 1970) at a 10-day time step. The NSE criterion represents the overall agreement of the shape of the hydrograph, while placing more emphasis on high flows. NSE values vary from -∞ to 1 with a maximum score of 1 meaning a perfect agreement between the observed and simulated values. At the contrary, negative values mean that more realistic estimates are obtained using the observed mean values rather than the simulated ones.

$$NSE = 1 - \left\{ \frac{\sum_{t=1}^{N}(Q_{obs}^t - Q_{sim}^t)^2}{\sum_{t=1}^{N}(Q_{obs}^t - \overline{Q_{sim}^t})^2} \right\} \tag{6}$$

where *NSE* Nash-Sutcliffe efficiency, $Q_{obs}^t$ and $Q_{sim}^t$ are, respectively, the observed and simulated streamflow for time step *t*; and *N* is the number of time steps for which observations are available.

The distribution of the basin in the study region (Fig. 1) provided the opportunity to assess the hydrological consistency of the SPPs in space by running GR4j in the four basins over the common 2008–2012 period of observed discharge availability. The hydrological consistency of the SPPs over time was then evaluated by running the model over the entire 2000–2012 period for which discharge observations were only available in two catchments (Katari and Keka). For each precipitation input ($P_{ref}$ and SPPs), the model was calibrated against observed streamflow over the whole period (2000–2012) and over three 4-year sub-periods (2000–2004, 2004–2008, and 2008–2012). No validation step was used as the objective was to assess hydrological modelling sensitivity to various precipitation datasets ($P_{ref}$ and SPPs) and not to assess the hydrological model robustness under

climate variability. The aim of the analysis of the streamflow simulation accuracy among the basins and periods considered was thus to evaluate the strength of the SPPs in space and over time and discrepancies in reproducing streamflow.

### 3.3 SPPs as input data for snow modelling in the Andes

A distributed, degree-day model (Ruelland et al., submitted) was chosen to analyse the sensitivity of snow cover simulations to the SPPs. This storage-based model relies on daily, distributed precipitation (P), temperature (T) and potential evaporation (PE) (Eq.7) to represent the main snow accumulation and ablation (sublimation and melt) processes (see Fig. 4). It operates at a daily time step according to a grid of 500 x 500 m corresponding to the spatial resolution of the MODIS data.

Snow accumulation is defined using a temperature threshold $Ts$ fixed at 0°C. Sublimation is accounted for based on daily $PE_{sub}$ (mm) at the target grid-cell and snowmelt is controlled with a melt factor parameter $Kf$ (°C$^{-1}$d$^{-1}$ to be calibrated) according to Eq. (8-9).

$$PE_{sub} = PE \times K_{sub} \tag{7}$$

where $PE_{sub}$ is potential evapo-sublimation, $PE$ is potential evaporation (see Eq. 5); and $K_{sub}$ is a proportional coefficient depending on the mean latitude (lat, decimal degrees) of the study area and varying from 0 to the poles to 1 at the equator (see Ruelland et al., submitted for more details)

$$Mf = Kf \times (T - Ts) \tag{8}$$

$$Melt = \begin{cases} 0 & T_j \leq Ts \\ Min\,(SWE, Mf) & T_j > Ts \end{cases}; \tag{9}$$

where $Mf$ is the potential melt (mm), $Kf$ is a melt factor parameter to be calibrated, $T$ is the temperature on day $j$ on the grid-cell considered, $Ts$ is the threshold temperature parameter (fixed at 0°C) for snow accumulation and melt. $Melt$ cannot exceed the snow water equivalent ($SWE$) of the snowpack storage.

The snow-covered areas (SCA) are estimated from the $SWE$. For each grid-cell, snow is stored in a reservoir which represents the $SWE$ of the grid-cell snowpack (see Fig. 4). It is fed solely by the solid fraction of precipitation and is emptied according to the simulated sublimation and melt processes. For each model, a grid-cell is assigned to snow or not depending on a water level threshold $SWE\_th$ (mm), to be calibrated.

As the region contains permanent snow covered areas, for grid-cells located below and above 5,700 m asl, the $SWE$ reservoir was initialized to 0 mm and 300 mm, respectively, at the beginning of the simulations. These values were defined based on MODIS snow observations and on model sensitivity tests to SWE initial conditions accounting for different elevation thresholds. The analysis revealed limited sensitivity to initial conditions (data not shown). An initial 3-year warm-up period was used for each simulation to limit the influence of these conditions.

The following ranges were used for model calibration with all the tested precipitation datasets ($P_{ref}$ and SPPs): 0.5 mm.°C$^{-1}$d$^{-1}$ < $Kf$ < 20 mm.°C$^{-1}$d$^{-1}$, 1 mm < $SWE\_th$ < 80 mm. $Kf$ ranges were based on ranges adapted from the values reviewed in Hock (2003). Regarding the $SWE\_th$ value, the assumption is that, using remote sensing, snow cover cannot be detected below a certain threshold (Bergeron et al., 2014). For instance, based on in-situ measurements to detect snow cover from MODIS in the Pyrenees, Gascoin et al., (2015) found a mean threshold of 40 mm. Since this value may be influenced by the local context (vegetation, topography, and climate) and spatial difference between point (in-situ) and areal satellite (MODIS) observations, $SWE\_th$ was tested according to large ranges. It is worth mentioning that the tested bounds were reached during calibration for all simulations (i.e. $Kf$=20 mm°C$^{-1}$d$^{-1}$ and $SWE\_th$=1 mm). However, we did not consider larger parameter ranges to ensure "realistic" simulations.

In association with $T_{ref}$ for temperature forcing data (see section 2.2.), $P_{ref}$ and SPPs were sequentially used as forcing precipitation datasets to simulate snow cover with the model. For each run, model parameters were calibrated based on the shuffled complex evolution (SCE) algorithm by optimising the grid-cell-to-grid-cell correlation between the snow cover duration (SCD) simulated by the model and that observed by the gap-filled MODIS snow products (see section 2.3.2.), according to the following Eq. (10):

$$R^2 = 1 - \frac{\sum_{p=1}^{n}\left(SCD_{MODIS(p)}-SCD_{MODEL(p)}\right)^2}{\sum_{p=1}^{n}\left(SCD_{MODIS(p)}\right)^2} ; \tag{10}$$

where $R^2$ is a determination coefficient, $n$ is the total number of grid-cells in the study area (see Fig. 1); $p$ is a given grid-cell; $SCD_{MODIS}$ and $SCD_{MODEL}$ are, respectively, the snow cover duration (SCD) observed by MODIS and the SCD simulated by the model as a percentage of days over the analysis period.

For each precipitation input ($P_{ref}$ and SPP), the model was calibrated against MODIS observed snow cover over the entire period 2000–2012 and over three 4-year sub-periods (2000–2004, 2004–2008, and 2008–2012). The aim of the analysis of the snow simulation accuracy among the areas and periods considered was to evaluate the strength of SPPs in space and over time and to identify discrepancies in reproducing snow cover in a remote Andean area (see Fig. 1).

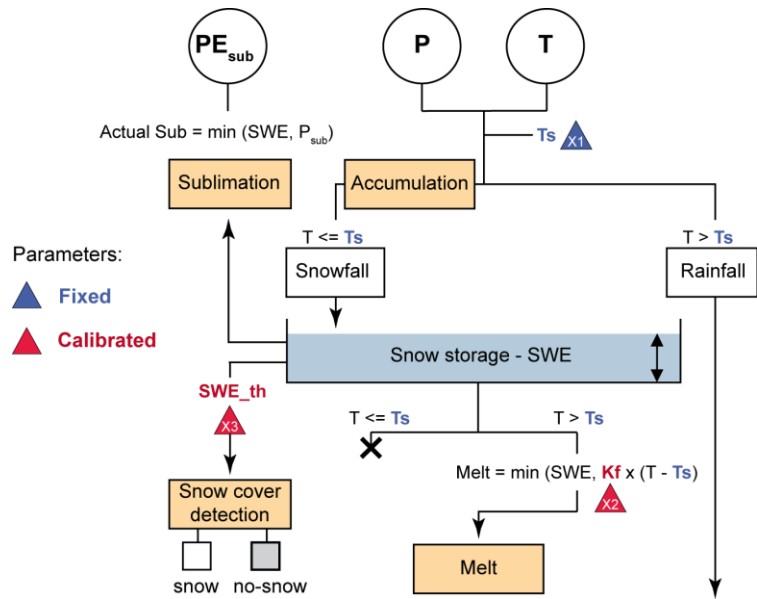

**Figure 4: Distributed, degree-day model used in the study (Ruelland et al., submitted).**

## 4. Results

### 4.1. Space-time consistency of SPPs compared with gauge observations

SPPs are distributed over the Taylor diagram, indicating high discrepancy among them for the 2000-2012 period (Fig. 5a). Some SPPs greatly overestimated precipitation with %B values of 165%, 75%, 40% and 45% for CMORPH–CRT v.1, TMPA–RT v.7, PERSIANN and PERSIANN–CDR, respectively. CMORPH–RAW v.1, GSMaP v.6 and SM2Rain–CCI v.2, greatly underestimated precipitation with %B values of -24%, -25% and -37%, respectively. However, all SPPs were highly correlated with $P_{ref}$ with CC greater than 0.75. Generally, including gauge data in the SPP processing clearly enhanced precipitation estimates. Indeed, CHIRPS v.2, CMORPH-BLD v.1, PERSIANN–CDR and TMPA-Adj v.7 were closer to the reference dot than their respective non-adjusted version, CHIRP v.2, CMORPH–RAW v.1, PERSIANN and TMPA-RT v.7. With the closest and farthest distance to the reference, MSWEP v.2.1 and CMORPH–CRT v.1 were respectively the most and least consistent SPPs to represent the mean spatially averaged precipitation over the 2000–2012 period.

According to the literature, a quality threshold value can be used to express SPP potential. Some authors (e.g., Hussain et al., 2017; Satgé et al., 2016; Shrestha et al., 2017), considered that a normalized RMSE value lower than 0.5 is associated with a very good SPP performance. Even there were slight differences between CRMSE and RMSE, the use of a normalized CRMSE threshold value of 0.5 to select only the most efficient SPPs remains logical. Therefore, six SPPs were selected for the following assessment steps including CHIRP v.2, CHIRPS v.2, CMORPH–BLD v.1, PERSIANN–CDR, TMPA–Adj v.7, and MSWEP v.2.1.

At the regional scale, SPP rank performance remained stable during the time windows considered and similar to what was observed for the 2000–2012 period (Fig. 5). Therefore, at the regional scale, SPPs were generally consistent over time, MSWEP v.2.1 being the most accurate and PERSIANN-CDR the least accurate SPP. However, for the 2008–2012 period, CHIRPS v.2 and CHIRP v.2 were closer than for the previously considered period. This might be due to the decrease in the number of available gauges for the adjustment processes applied on CHIRP v.2 to produce CHIRPS v.2.

Figure 5c shows the spatial distribution of SPP errors for the 2000–2012 period in terms of %B, CC, and CRMSE. CMORPH–BLD v.1 was poorly correlated with $P_{ref}$ with the highest proportion of grid-cells with CC less than 0.7. This value is generally used as a quality threshold with CC values less than 0.7 indicating poor SPP performance (see e.g., Satgé et al., 2016). MSWEP v.2.1 had the best CC value overall with the highest proportion of grid-cells with a CC greater than 0.9 and only two grid-cells with unsatisfactory CC values.

The main differences were in %B and CRMSE. CMORPH-BLD v.1 and PERSIANN-CDR precipitation under- and over-estimations at the regional scale (Fig. 5a) were confirmed at the gauge scale (Fig. 5c). CHIRP v.2, CHIRPS v.2, and TMPA–Adj v.7 presented similar %B distribution while MSWEP v.2.1 had the most homogeneous %B distribution with the values of almost all grid-cells ranging between -30% and +30%. This range was previously defined as satisfactory %B for SPPs (Shrestha et al., 2017). The gauge adjustment applied on CHIRP v.2 was globally positive with a %B reduction from CHIRP v.2 (15.9%) to CHIRPS v.2 (0.4%) of almost 100% (Fig. 4a). It considerably increased the numbers of grid-cells with %B between -15% and +15% from CHIRP v.2 to CHIRPS v.2 (Fig. 5b) and generally enhanced CRMSE and CC scores.

Interestingly, all the SPPs underestimated precipitation for the two grid-cells located over the northern Lake Titicaca islands. This is probably linked to SPP's limited ability to detect warm cloud precipitation (see section 5.1).

CMORPH–BLD v.1, PERSIANN–CDR, and TMPA–Adj v.7 had the highest proportion of grid-cells with CRMSE values greater than 0.7 (Fig. 5c). This value can be used as a quality threshold above which SPPs performance is considered as unsatisfactory (see e.g. Shrestha et al., 2017). Therefore, over the 2000–2012 period, precipitation estimates derived from CMORPH–BLD v.1, PERSIANN–CDR and TMPA–Adj v.7 are subject to high uncertainties at local scale. The inclusion of gauge observations for CHIRPS v.2 estimates reduced the number of grid-cells with unsatisfactory performance by 50% in comparison with the non-adjusted CHIRP v.2 version. With only 14 and 12 grid-cells with CRMSE greater than 0.7, respectively, CHIRPS v.2 and MSWEP v.2.1 showed the highest spatial consistency.

More generally, the spatial analysis highlighted three areas in which all the SPPs considered presented less satisfactory statistical scores (CRMSE, CC, and %B) than over the remaining areas. These regions correspond to the south eastern Lake Titicaca shore, and the south-western and north-eastern borders of the catchment.

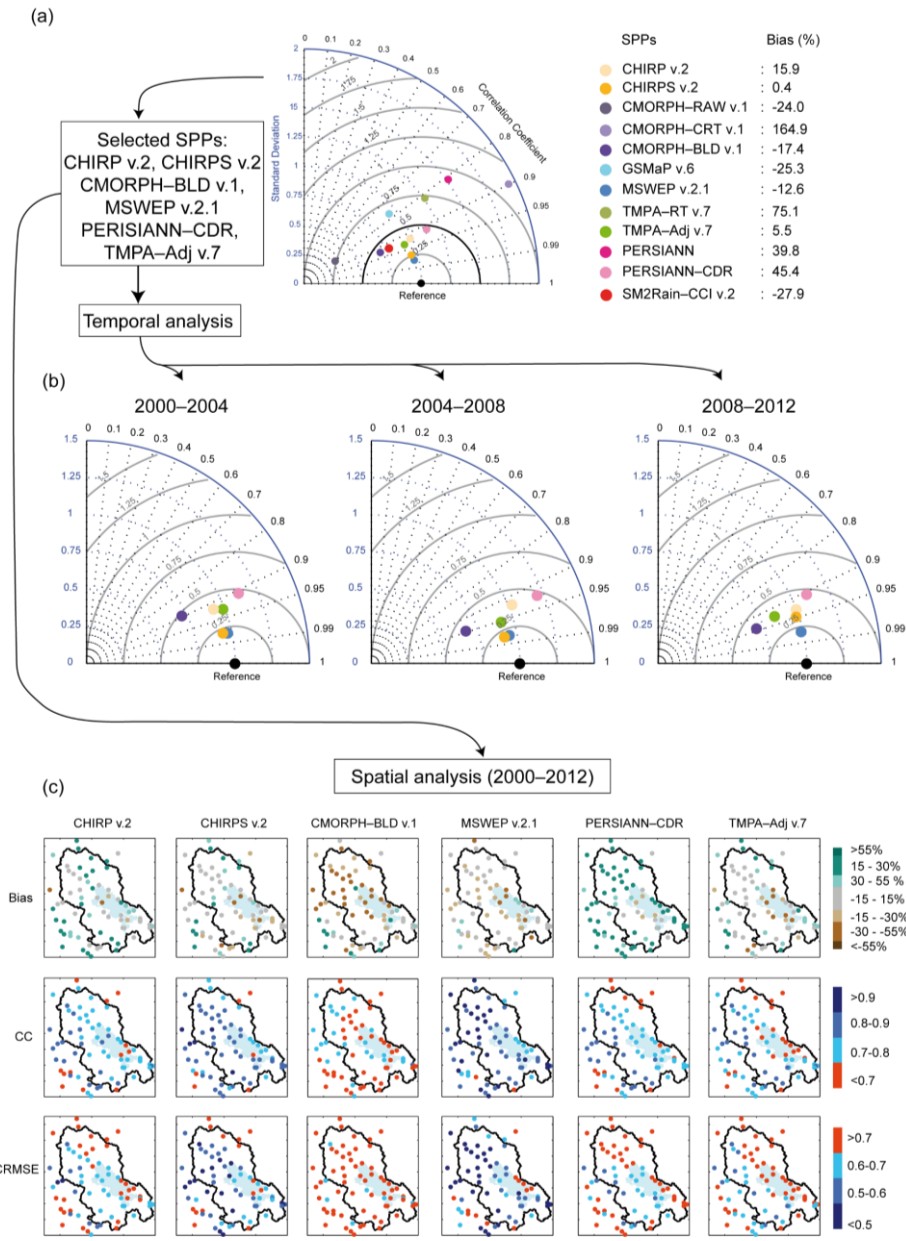

**Figure 5: Efficiency of SPPs compared with gauge observations: (a) Pre-selection of SPPs at the regional scale in the form of a Taylor diagram; (b) Consistency of SPPs over time at the regional scale in the form of a Taylor diagram; and (c) Consistency of SPPs in space for the 2000–2012 period from CC, CRMSE and %B values obtained from each $P_{ref}$ grid-cell including at least one gauge.**

The SPPs' lowest potential over the south-eastern shore of Lake Titicaca and at the south-western and north-eastern borders of the catchment highlighted in the 2000–2012 period was confirmed for all four time windows, as indicated by CRMSE values (Figure 6). For each time window, the CRMSE distribution differed depending on the SPP considered. More

interestingly, the distribution of CRMSE of all the SPPs differed depending on the time window considered. As a result, the efficiency of the SPPs varied over time for a specific region (at the grid-cell scale). As a clear example, for the CMORPH–BLD v.1 south-eastern located grid-cells, the CRMSE values changed drastically over time: these grid-cells presented unsatisfactory CRMSE scores (above 0.7) for the 2000–2004 period and satisfactory CRMSE scores (below 0.5) for both the 2004–2008 and the 2008–2012 periods.

In this context, Fig. 6d represents the space-time consistency of the SPPs over the 2000–2012 period. The mean and STD CRMSE values obtained at the location of each grid-cell in the three sub-periods considered, used as indicators of the space-time consistency, are plotted in Fig. 6d. Overall, all SPPs are stable in space and over time with STD CRMSE values below 0.25 but their consistency differed in accuracy. CMORPH–BLD v.1 provided stable but not accurate precipitation estimates with mean CRMSE values systematically above 0.7. In contrast, MSWEP v.2.1 and CHIRPS v.2 presented stable and accurate precipitation estimates with many grid-cells with a mean CRMSE below 0.5. CHIRP v.2 space-time consistency was lower than that of CHIRPS v.2, thereby confirming the advantage of the gauge calibration (Fig. 6d). PERSIANN–CDR and TMPA–Adj v.7 were more consistent in space and over time over the southern mid and western region, respectively. Interestingly, the close potential precipitation estimates observed for CHIRP and CHIRPS v.2 at the regional scale and for the 2008–2012 period were confirmed at the grid-cell level with similar spatial error distribution for both SPPs. Therefore, gauge adjustment appears to be less efficient for the 2008–2012 than for the 2000–2004 and 2004–2008 periods.

MSWEP v.2.1 and CHIRPS v.2 were the most stable SPPs in space and over time with the highest proportion of grid-cells with a mean and a STD CRMSE below 0.5 and 0.25, respectively (Fig. 6d). However, for  grid-cells  located  on  the south-eastern shore of the lake, CHIRPS v.2 provided more accurate and stable precipitation estimates than MSWEP and all the other SPPs.

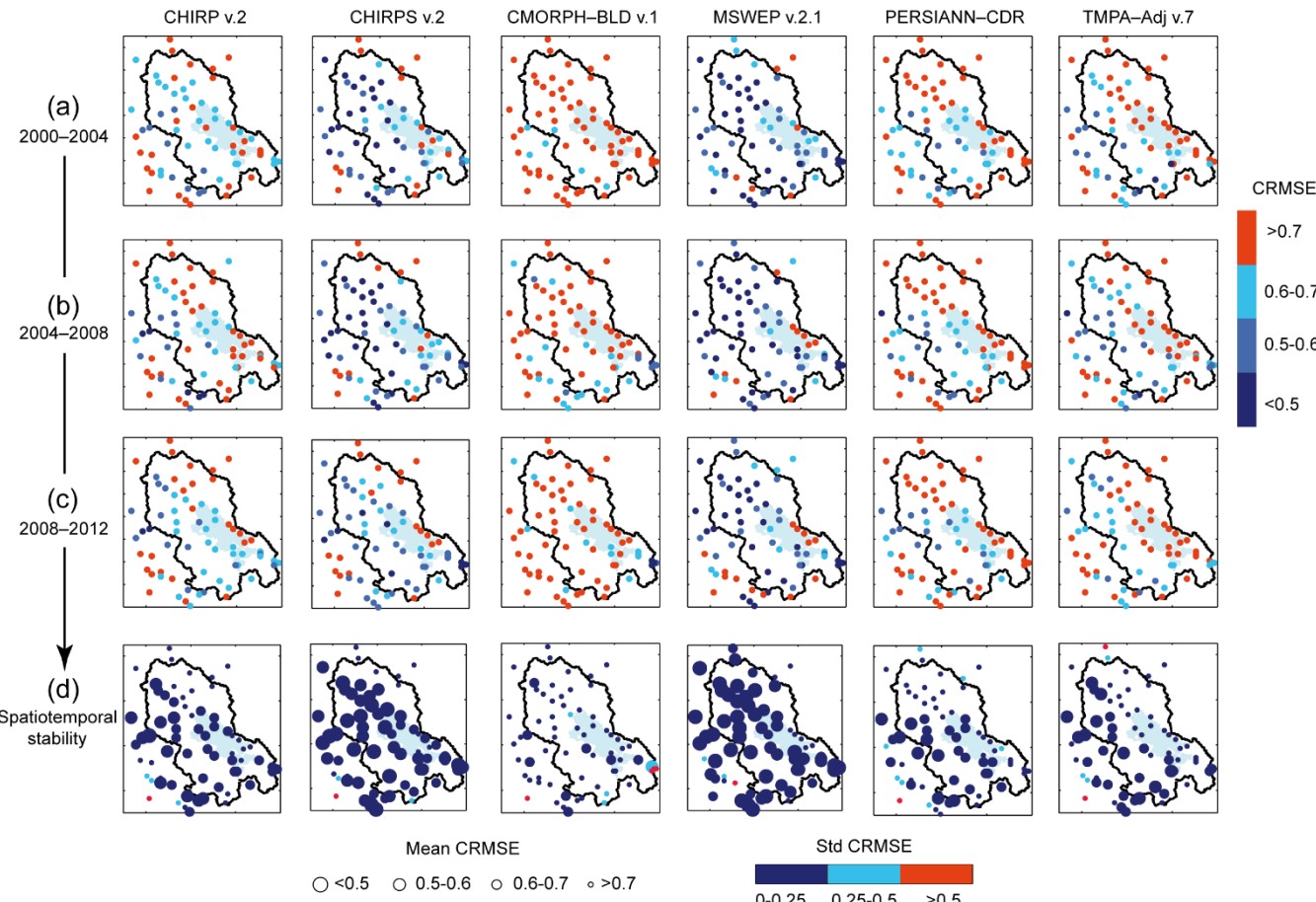

**Figure 6: Consistency of SPPs in space and over time compared with gauge observations. CRMSE values obtained from the SPPs at the gauge level for three 4–year periods: (a) 2000–2004; (b) 2004–2008; and (c) 2008–2012. (d) Mean and STD CRMSE values obtained for each grid-cell using the three respective values obtained for the three 4–year periods.**

## 4.2. Space-time consistency of SPPs compared with streamflow simulations

Streamflow simulations using $P_{ref}$ varied along the catchments with lowest NSE score of 0.63 for Keka and the highest of 0.89 for Katari. Simulated SPP streamflow efficiency followed the same trend except with CMORPH–BLD v.1 and TMPA–Adj v.7 (Fig. 7). TMPA–Adj v.7 provided the best streamflow simulation for the Ilave catchment and the worst for Keka, whereas the opposite was observed using CMORPH–BLD v.1 as forcing data. The low efficiency of CMORPH–BLD v.1 in the Ilave catchment was related to its erroneous streamflow peaks in the 2009 and 2012 dry seasons. CHIRP v.2 and CMORPH–BLD v.1 had the lowest scores for the Katari and Ramis catchments with NSE values of 0.59 and 0.45, respectively. For all the catchments, streamflow simulations based on CHIRPS v.2 presented systematically higher NSE scores than simulations based on CHIRP v.2, showing that the adjustment provided in CHIRPS v.2 with the integration of gauge observations led to better precipitation estimates. This confirms the enhancement of the gauge-based assessment observed with an overall reduction in bias and an increase in CRMSE and CC.

For all the catchments considered, the best streamflow simulations were obtained with at least one of the SPPs as forcing precipitation data. TMPA–Adj v.7 and CMORPH–BLD v.1 provided a better streamflow simulation than $P_{ref}$ for Ilave and Keka respectively, and MSWEP v.2.1 outperformed $P_{ref}$ for the Katari and Ramis catchments. These results show that SPPs can efficiently replace the currently available sparse precipitation gauge networks for use in hydrological studies of the region. Overall, MSWEP v.2.1 appears as the most consistent SPP product for streamflow simulations. Indeed, the streamflow simulations forced by MSWEP v.2.1 were more realistic than those forced by $P_{ref}$ over three catchments (Katari, Keka, and Ramis) and were almost the same for the Ilave catchment.

However, as shown in Fig. 7c, the SPP hydrological ranking in the 2008–2012 period changed drastically over time. For example, for the Katari catchment, MSWEP v2.1 led to the best streamflow simulations for the 2004–2008 and 2000–2012 but not for the 2000–2004 period, for which TMPA–Adj v.7 forced streamflow simulations had a higher NSE score of 0.85. Additionally, CMORPH–BLD v.1 potential fell drastically over the 2000–2004 period with a negative NSE score, whereas it produced the most realistic streamflow simulation for the period 2008–2012. In the Keka catchment, for each time window, the best streamflow simulation was obtained using different SPPs. CHIRP v.2, PERSIANN–CDR and CMORPH–BLD v.1 resulted in the highest NSE scores over the various sub-periods analysed, with respectively 0.73 for the period 2000–2004, 0.83 for 2004–2008 and 0.77 for 2008–2012.

To conclude, the analysis of streamflow model sensitivity to the different precipitation datasets depended to a great extent on the time window and catchment considered. However, MSWEP v.2.1 appeared to be the most 'stable' SPP. It provided the most realistic streamflow simulations with higher NSE scores than $P_{ref}$ for seven out of ten simulations and also outperformed other SPPs in almost all streamflow simulations tested. The results obtained for the 2000–2012 period faithfully reflected global SPP performance for all the time windows with MSWEP v.2.1 as the most and CMORPH-BLD v.1 as the least suitable SPP for hydrological modelling over the Katari and Keka catchments.

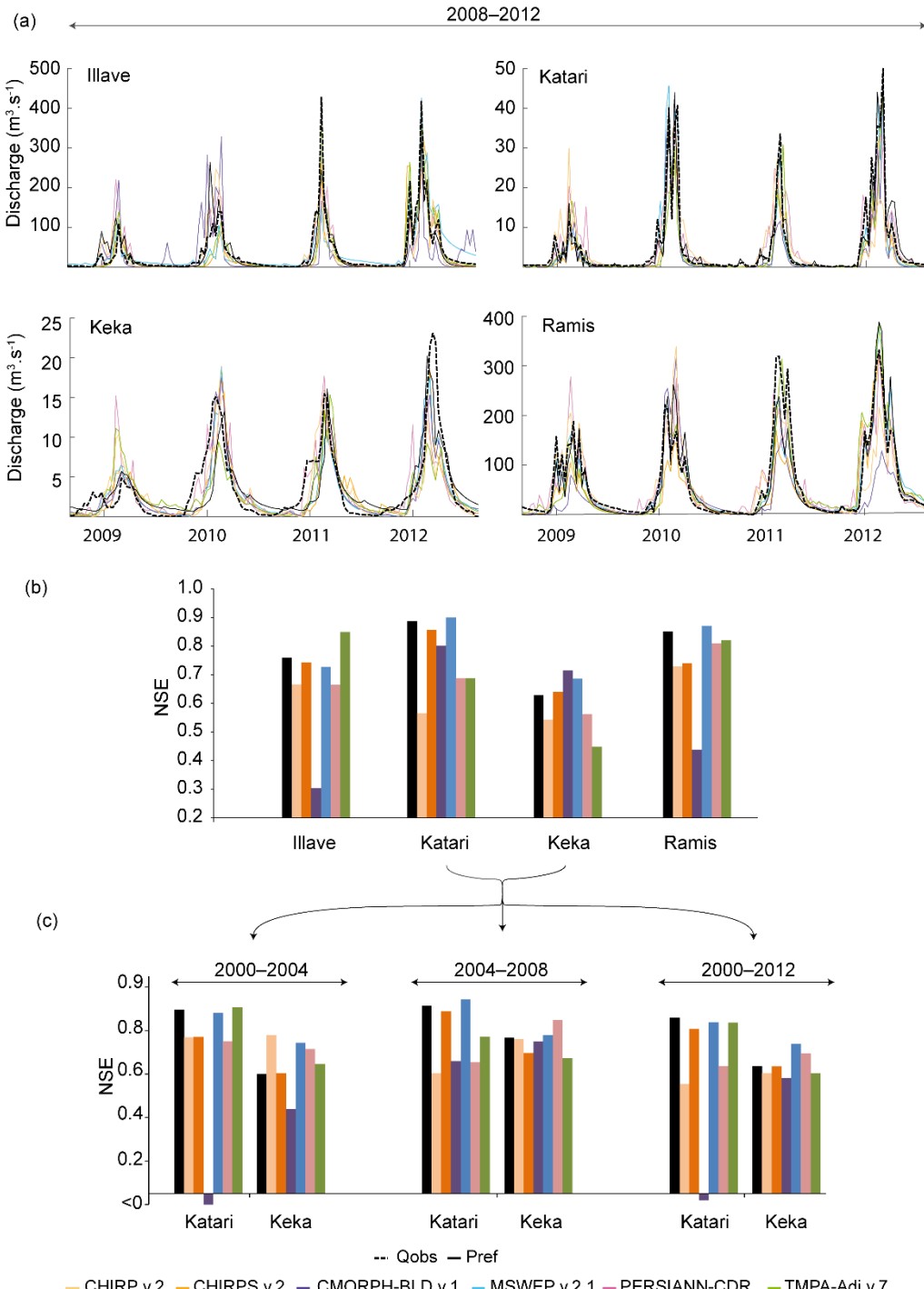

**Figure 7: Observed versus simulated streamflow using $P_{ref}$ and SPPs as input data in the hydrological model: (a) 10-day streamflow simulations at the four basin outlets (Ilave, Katari, Keka and Ramis) over the 2008–2012 calibration period; (b) corresponding efficiency (NSE scores) of simulated versus observed streamflow; (c) efficiency (NSE score) of simulated versus observed streamflow over the 2004–2004, 2004–2008 and 2000–2012 calibration periods at two basin outlets (Katari and Keka).**

## 4.3. Space-time consistency of SPPs for snow cover duration

Whatever the period, the best SCD simulations were obtained using $P_{ref}$ as input data with $R^2$ values of 0.83, 0.80, 0.79 and 0.82 for the 2000–2004, 2004–2008, 2008–2012 and 2000–2012 periods, respectively (Fig. 8). Simulated SCD using SPPs as input data systematically overestimated SCD compared to the SCD computed from the gap-filled MODIS snow products. The MSWEP v.2.1 (CHIRPS v.2) datasets provided the most realistic (unrealistic) SCD simulations over 2000–2012 with R² of 0.80 and 0.67 respectively (see Fig. 8a). Interestingly, mean annual precipitation over 2000–2012 (see Fig. 8a) was significantly higher with the SPPs (ranging from 762 mm with MSWEP v.2.1 to 1,229 mm with CHIRP v.2) than with $P_{ref}$ (523 mm). The least realistic SCD simulations with the SPPs may thus be explained by higher precipitation, which increases snowfall input and snow cover duration despite the specific calibration of the snow model. Indeed, MSWEP v.2.1 mean annual precipitation (762 mm) was the closest to the $P_{ref}$ ones and provided the most realistic SCD estimates. Conversely, CHIRP v.2 provided the highest mean annual precipitation estimate (1,229 mm) whereas it produced the least realistic SCD estimates with $R^2$ values of 0.74, 0.58, 0.65, and 0.67, for the 2000–2004, 2004–2008, 2008–2012 and 2000–2012 periods, respectively. This trend was observed for all the periods considered (see Fig. 8b). All the other SPPs presented relatively close mean annual precipitation estimates and therefore relatively close SCD simulation performances (Fig. 8b).

Interestingly, all the SCD simulations based on SPPs as inputs had higher scores for the period 2000–2004 with minimum efficiency for CHIRP v.2 ($R^2$=0.74) and maximum for MSWEP v.2.1 ($R^2$=0.83). This is not in line with the space and time analysis based on gauges and hydrological modelling analyses. Indeed, CMORPH-BLD v.1 was the least efficient SPP to represent $P_{ref}$ and observed streamflow over 2000–2004, while its SCD estimates was the second most efficient SCD simulation forced by SPPs. Similarly, the CHIRP v.2 forced streamflow simulation was the most realistic one in the Keka basin for the period 2000–2004, while the CHIRP v.2 SCD simulation was the least efficient in comparison to the other SPPs. The TMPA–Adj v.7 forced streamflow simulation was the first most realistic one for the Ilave basin and the second most realistic one for the Ramis basin over the period 2008–2012, whereas its SCD simulations were among the least efficient ($R^2$=0.67) in comparison to the other SPPs. This shows that a SPPs can be effective for a specific region or for a given indicator (i.e. gauges and hydrological modelling) but not for another region or indicator (i.e. snow modelling) and inversely. However, the analysis also confirmed the MSWEP v.2.1 overall stability and efficiency in space and over time previously highlighted for the precipitation and streamflow representation. Indeed, MSWEP v.2.1 forced SCD simulations remained relatively stable with similar scores obtained for all the periods considered.

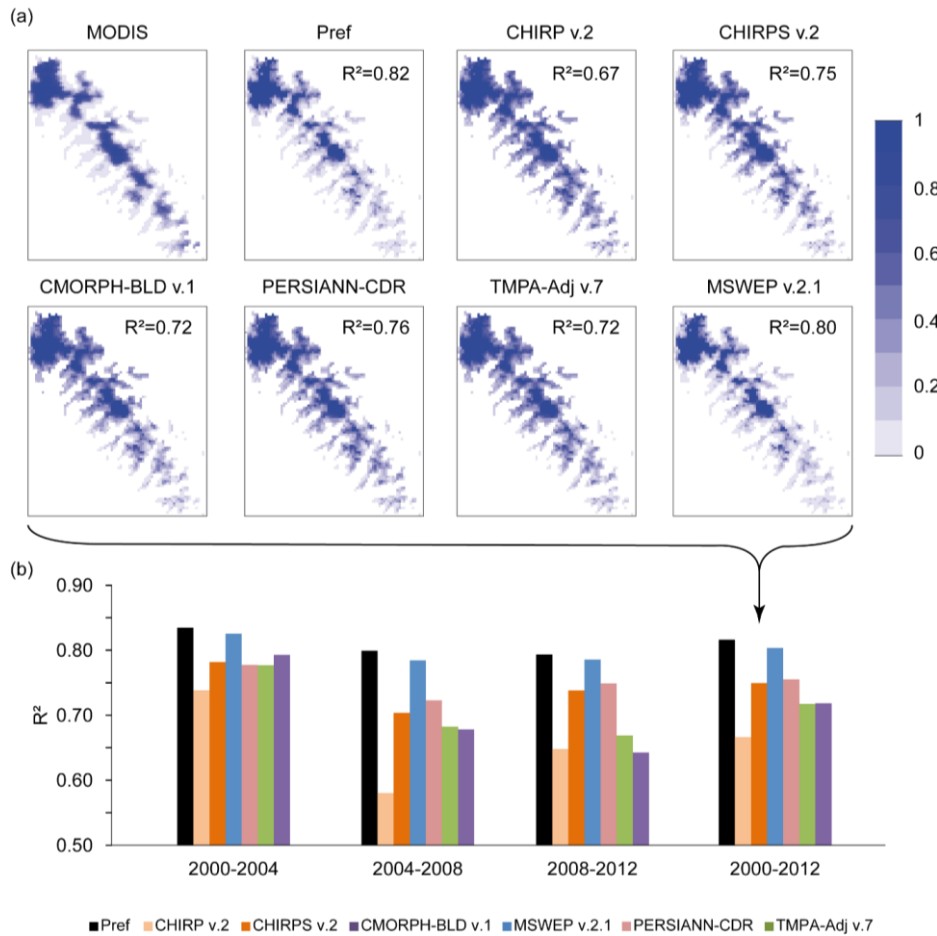

**Figure 8: Observed (gap-filled MODIS snow-products) versus simulated snow cover duration (SCD) using $P_{ref}$ and SPPs as input data in the snow model: (a) Maps of snow cover duration for 2000–2012; (b) Efficiency ($R^2$ scores) of simulated SCD versus MODIS SCD for the different calibration periods. SCD are expressed as the percentage of days over the simulation period. $R^2$ values stand for the grid-cell-to-grid-cell correlation between the observed and simulated SCD. Green and red stars highlight SPPs with closest and further precipitation estimates to $P_{ref}$ over the subset considered.**

## 5. Discussion and conclusion

### 5.1. SPP spatial variability: the Lake Titicaca bias

At the regional scale, the performances of the SPPs were in relatively good agreement with the gauge references. However, their performances differed markedly in space with a general trend to underestimating precipitation over Lake Titicaca.

Due to its size, the solar radiation absorption capacity of Lake Titicaca increases the temperature by 4° to 6° over the superficial water layer in comparison to that over the surrounding land (Delclaux et al., 2007; Roche et al., 1992). Additionally, evaporation from Lake Titicaca is very high, estimated at 1,700 mm.year[-1] (Pillco Zolá et al., 2018). Therefore, crossing the lake, the air masses pick up lake moisture which increases their temperature and allows their ascension. This convection results

in more precipitation over the lake than over the surrounding land (Roche et al., 1992). These precipitation events originate from warm clouds whose detection remains challenging for SPPs. Indeed, SPPs use cloud top IR temperature to discretize rainy and rainless clouds. The IR temperature threshold may be too low to correctly detect warm rain clouds, as suggested over mountainous region (see e.g., Dinku et al., 2010, 2007; Gebregiorgis and Hossain, 2013; Hirpa et al., 2010; Li et al., 2013).

Consequently, many precipitation events may be lost leading to underestimation of precipitation over Lake Titicaca. To support our hypothesis, we computed the probability of detection (POD) of daily precipitation events. POD is an indicator of a SPP's ability to correctly forecast precipitation events with values ranging between 0 and 1 and a perfect score of 1. A low POD indicates that precipitation events are not detected by SPPs. Figure 9 shows the POD obtained from all SPPs for each grid-cell with gauges over the 2000–2012 period. With relatively lower POD values for grid-cells above the lake than for grid-cells over the surrounding land, TMPA–Adj v.7 and CHIRPS v.2 exhibited a clear trend to underestimate daily precipitation occurrence over the lake. MSWEP v.2.1 and CHIRP v.2 showed the same trend over the northern part of the lake. These POD "anomalies" could partially explain the negative bias observed over the Lake Titicaca because of the warm cloud precipitation process. CMORPH–BLD v.1 and PERSIANN–CDR showed no significant trends and were the least efficient SPPs overall.

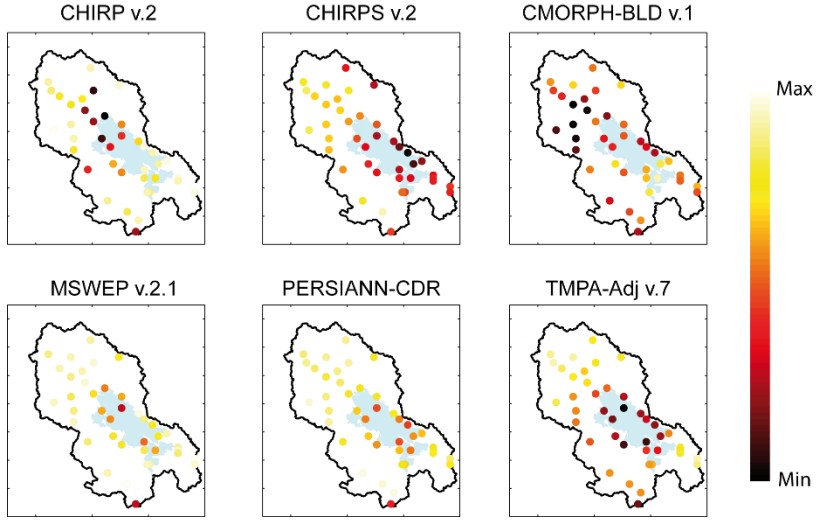

**Figure 9: SPP's ability to forecast daily precipitation events represented in the form of POD. POD=a/a+b with "a" being the number of days on which both SPP and P$_{ref}$ detected precipitation and "b" the number of days on which only P$_{ref}$ detected precipitation events. The legend shows the range between min and max to focus on POD anomalies over Lake Titicaca. Grid-cells located outside the Lake Titicaca basin were removed to facilitate interpretation.**

It is noteworthy that the eastern lake border is characterized by significant emissivity and temperature changes (induced by the lake and by the snow cover in the Cordillera), which perturb PMW precipitation retrieval (see e.g., Ferraro et al., 1998; Levizzani et al., 2002; Tian and Peters-Lidard, 2007) and may also contribute to the SPPs relatively lower performance.

River streamflow and precipitation over the lake surface account for respectively, 53% and 47% of the total Titicaca water supply (Roche et al., 1992). Even if MSWEP v.2.1 led to realistic simulations of streamflow, its absolute bias greater than 35%

over the lake is a major problem when attempting to model the Lake Titicaca water budget. Therefore, MSWEP v.2.1 precipitation estimates need to be adjusted over the lake. Previous studies merging gauges and SPPs successfully applied over the Altiplano (Blacutt et al., 2015; Heidinger et al., 2012; Vila et al., 2009) and elsewhere (Ma et al., 2017; Ringard et al., 2017) should be used as a guideline.

## 5.2. Gauges versus hydrological modelling-based assessment

Most of the studies on comparisons of SPP potential rely more on gauge-based assessment (see reviews by Maggioni et al., 2016 and Sun et al., 2017) rather than on hydrological modelling based assessment (see review by Maggioni and Massari, 2018). However, as shown in this study, the SPP's performance compared with gauge observations is not systematically supported by the sensitivity analysis of streamflow modelling. To provide more insight into this discrepancy, Fig. 10 compares SPP performance with precipitation gauges and with observed streamflow at the outlet of the four basins considered (Ilave, Katari, Keka, and Ramis). To simplify the analysis, the comparisons focus on the best and worst SPPs obtained from both gauges and hydrological analysis. Over the Katari and Ramis basins, MSWEP v.2.1 was the most efficient SPP, according to both gauges and hydrological assessment. However, for the Ilave and Keka basins, the most efficient SPPs varied with the assessment indicator. For the Ilave basin, MSWEP v2.1 was the most efficient SPP in representing precipitation at the gauge level while TMPA–Adj v.7 provided the best streamflow simulation. For the Keka basin, CHIRPS v.2 and CMORPH–BLD v.1 were the most efficient SPPs according to the gauges and hydrological analysis, respectively. Similar discrepancies were also observed concerning the least efficient SPPs. Indeed, for the Katari, Keka and Ramis basins, the least efficient SPPs varied with the assessment indicator considered (i.e. gauges or hydrological). We identified two main factors to explain those discrepancies.

The first factor is the spatial distribution of precipitation gauges in the catchment. For a given basin, the observed outlet streamflow includes precipitation over the basin. Therefore, unlike the gauge-based assessment, the hydrological assessment allows indirect feedback from the SPP's ungauged grid-cells in the basin. The ratio of "suitable" to "unsuitable" SPP grid-cells at the basin scale may differ from the point gauge assessment of SPP grid-cells and explain part of the discrepancy between the gauges and hydrological-based assessment. This difference could be even more marked when the gauges are located in specific grid-cells where SPPs do not provide efficient precipitation estimates due to geomorphological context or to spurious SPP anomalies, as shown by Chen and Li, (2016) and Satgé et al. (2017b). The hydrological-based assessment makes it possible to overcome these problems (distribution and density of the gauges). Besides, errors in the spatial distribution of precipitation can be partially offset by model calibration. Except when a dense network of precipitation gauges adequately captures the precipitation spatial patterns, hydrological-based evaluation appears to be a necessary step to complement SPP gauge-based assessments.

The second factor is the difference in spatial representation between point (gauge) and areal (SPP grid-cell) measurement of precipitation. An aeral measurement (SPP grid-cell) can be considered as the aggregation of "infinity" point measurements

(gauges). "Infinity" is hard to represent with limited spatial gauge measurements. At the SPP grid-cell scale, precipitation events can occur in the vicinity of the gauges but not at the exact location of the gauges. Consequently, differences in precipitation estimates derived from the gauges and the SPPs are not only related to SPP deficiencies, but also to the difference in their respective measured spatial scale. As a result, as shown by Tang et al. (2018), the use of sparse gauge networks tends to underestimate SPP potential. This would be even more marked for lower spatial resolution SPPs (TMPA–Adj v.7, PERSIANN–CDR, and CMORPH–BLD v.1, in this study) than for higher spatial resolution SPPs (CHIRP v.2, CHIRPS v.2, and MSWEP v.2.1, in this study) as the difference in spatial representation between a gauge and a SPP is greater. Indeed, the gauge-based assessment ranked the CHIRPS v.2, MSWEP v.2.1 as the best SPPs for this study area. However, aggregation of precipitation at the basin scale eliminated the difference in spatial representation between point (gauge) and areal (SPP) precipitation as both gauge and SPP aimed to represent precipitation at the same spatial scale (basin scale). Therefore, unlike gauge precipitation measurements, streamflow measurements are not expected to be more representative of high than low spatial resolution SPPs. Indeed, as shown in Figure 10, the best performances are not systematically achieved with the highest SPP resolutions compared with streamflow observation. The results obtained over the Ilave and Keka basins illustrate this feature: over the Keka basin, CHIRPS v.2 (5 km) precipitation estimates were better whereas CMORPH–BLD v.1 (25 km) provided better streamflow estimates. Similarly, MSWEP v.2.1 (10 km) was more efficient than gauges, but TMPA–Adj v.7 (25 km) provided better streamflow simulation over the Ilave basin.

To summarize, SPP assessments based on a gauge network make it possible to detect very local SPP inconsistencies but are influenced by (i) the distribution of the gauges and (ii) the difference in spatial resolution (point vs. grid-cells). It is also worth mentioning that hydrological-based assessments may be influenced by the model itself. In this study, the SPP assessment relied notably on their ability to provide realistic streamflow simulations at the basin outlet using a lumped model. Distributed physical models could reinforce the potential assessment of each product based on spatial criteria (for instance humidity, water level), but this would be difficult over the Lake Titicaca region due to the scarcity of data. Similarly, other ET estimates could have been used, especially those based on remote sensing techniques to compensate for the limited availability of temperature gauge data. However, we would like to underline that our purpose was not to evaluate or discuss the model structure or ET estimates, but to highlight the complementarity of point gauges and integrated hydrological modelling. In this context, the same ET estimates and the same hydrological model were used to assess all the SPEs through the specific calibration of each dataset.

A complete quality assessment of SPPs is therefore difficult to achieve and the choice of the most suitable SPPs should rather be based on assessment steps and include the final use of the SPPs.

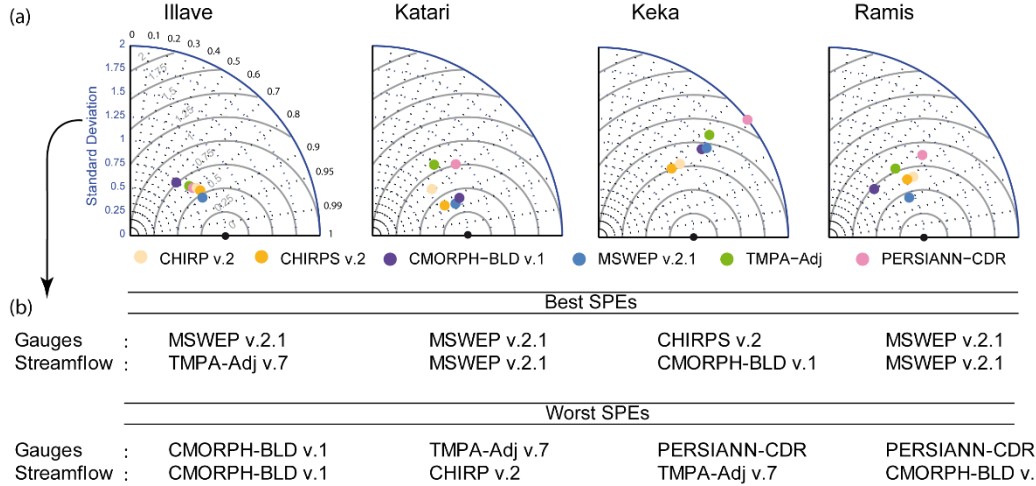

| | Illave | Katari | Keka | Ramis |
|---|---|---|---|---|

**Best SPEs**

| | | | | |
|---|---|---|---|---|
| Gauges : | MSWEP v.2.1 | MSWEP v.2.1 | CHIRPS v.2 | MSWEP v.2.1 |
| Streamflow : | TMPA-Adj v.7 | MSWEP v.2.1 | CMORPH-BLD v.1 | MSWEP v.2.1 |

**Worst SPEs**

| | | | | |
|---|---|---|---|---|
| Gauges : | CMORPH-BLD v.1 | TMPA-Adj v.7 | PERSIANN-CDR | PERSIANN-CDR |
| Streamflow : | CMORPH-BLD v.1 | CHIRP v.2 | TMPA-Adj v.7 | CMORPH-BLD v.1 |

**Figure 10. Comparison of the performance of the SPPs with that of precipitation gauges and observed streamflow at the outlet of the four basins (Ilave, Katari, Keka, and Ramis): (a) SPP performance at the gauge level for the four catchments represented according to a Taylor diagram (scores were computed using only the grid-cells included in the catchments for the 2008–2012 period); (b) Best and worst SPPs according to the gauge-based and hydrological-based assessments for the 2008–2012 period.**

## 5.3. Space-time SPP consistency

For each assessment indicator considered (i.e. gauges, streamflow and snow cover duration), the performances of the SPPs varied depending on the time window considered. These variations can be easily observed when SPPs are used as forcing data for hydrological modelling, in which case the ranking of SPPs observed for each of the 2000-2004, 2004-2008, 2008-2012 and 2000-2012 time windows changed significantly (see Fig. 12b and 12c). Therefore, as shown in the analysis, a given SPPs can appear to be the best option to represent precipitation for one particular period but not for another. Additionally, for the same period, the SPP may be suitable for a specific regional subset but not for another as clearly shown at the gauges and hydrological based assessment (see Fig. 5 and 12b). This space-time inconsistency is of major concern in the current context of climate variability. Indeed, over remote regions, the scarcity of meteorological stations encourages scientists to use remote sensing data to understand precipitation variability and its contribution to meteorological, agricultural and hydrological droughts (see e.g Agutu et al., 2017; Arvor et al., 2017; Tan et al., 2017; Tao et al., 2016; Bayissa et al., 2017; Satgé et al., 2017a). Therefore, for these studies, a consistent analysis of the consistency of SPPs in space and over time is a pioneering step to select the most realistic SPPs. It should make it possible to foresee potential propagation of SPP inconsistencies in the studies to consistently weight the observed results. From our analysis, it will be recalled that some SPPs are relatively stable at regional scale (CHIRPS v.2, MSWEP v.2.1) and could thus be used to study regional precipitation patterns with a relatively high degree of confidence. For studies on the spatial variability of precipitation, MSWEP v.2.1 is the most suitable SPP for the study region as it is the most stable in space and over time. However, as discussed above, correction methods need to be taken into consideration to enhance precipitation estimates over Lake Titicaca. Moreover, care should be taken in the north-eastern and south-western regions corresponding to the transition from the Amazon to the Altiplano and the Altiplano to the

Pacific watershed zones, respectively (see Fig. 6d). For these specific regions, all the SPPs studied here represented the lowest space-time stability. These regions present very significant variations in elevation, which are known to interfere in SPPs (Ochoa et al., 2014; Satgé et al., 2017b) and may partially explain this local discrepancy.

## 5.4. Snow modelling to assess SPPs over unmonitored regions

SPPs are subject to high uncertainty in mountainous regions. Indeed the precipitation/no precipitation classification based on cloud top IR temperature generally fails because the temperature threshold used for the discretization process is too high. Currently, over high elevation mountainous regions, the top cloud temperature is generally lower than over flat regions resulting in many non-rainy cold clouds being misidentified as rainy clouds (Hussain et al., 2017; Satgé et al., 2017a). In addition, snow and ice cover appear to be similar to ice precipitation aloft in the scattering signal in microwave channels (Ferraro et al., 1998; Levizzani et al., 2002) leading to the misidentification of snow and ice cover with rainy clouds (Dinku et al., 2010; Hussain et al., 2017; Mourre et al., 2016). As an example, the inconstancy described in both IR and PMW precipitation recovery led to marked overestimation by SPPs over the Himalayan Pakistan region (Hussain et al., 2017, Satgé et al., 2018). However, the accessibility and handling difficulty over high elevation snow covered regions limits the availability of gauges and hence the assessment of SPPs. In this context, the proposed snow modelling-based assessment using snow cover observed by satellite imagery offers an alternative way to obtain initial feedback on the efficiency of SPPs over completely unmonitored regions. The present study shows that gap-filled MODIS snow-products can be used as reference to indirectly assess SPP efficiency and show great promise for the validation of SPPs in regions where distinguishing between rainfall and snowfall is still challenging. However, the distribution of SCD (Fig. 8a) indicates permanent snow-covered areas over which seasonal dynamics from optical imagery (MODIS) are difficult to capture as there is more variation in depth than in spatial extent. Consequently, the regional seasonal snow cycle captured from MODIS is weak and erratic (see Fig. 11). Preliminary modelling tests showed that the dynamics of these snow-covered areas were poorly simulated by the snow model when the $P_{ref}$ and SPP precipitation datasets were used.

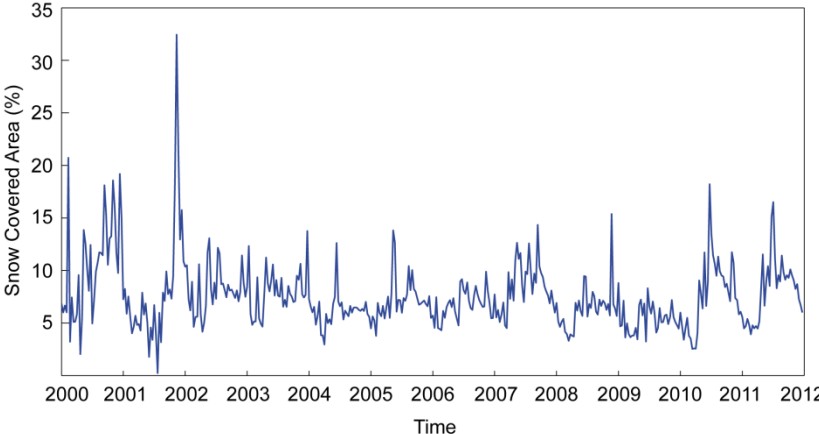

**Figure 11: Snow covered area observed from gap-filled MODIS snow products at a 10–day time step over the Andean subset.**

Another possible approach could consist of coupling satellite radar altimetry and optical imagery to monitor the seasonal snow cycle in terms of snow depth and volume. The recently launched Sentinel-2 (optical) and Sentinel-3 (altimeter) satellites present an unprecedented opportunity to monitor snow volumetric dynamics at finer spatial resolution than MODIS.

In addition, the SPPs tested in this study were not originally designed to distinguish between liquid and solid precipitation. The newly released SPPs Integrated Multi-satellite Retrievals for the Global Precipitation Mission (IMERG–v.3 v.4 v.5) are the first SPPs which make it possible to discretize liquid to solid precipitation at 10 km spatial resolution and almost global scale. We expect that such products will enhance the monitoring of seasonal snow dynamics. However, as yet, no studies have reported on the potential of IMERG products to estimate snowfall, which should guide the next step in the assessment of SPPs over the region.

## 6. Conclusions

This paper compared the space-time consistency of 12 SPPs at different spatial and temporal scales with point gauge observations and according to sensitivity analysis of snow-runoff responses. The main results of the study can be summarized as follows:

- Given the currently available precipitation gauge network, SPPs are attractive and efficient tools to monitor local precipitation and to force impact modelling, such as snow-hydrological models. Of the SPPs with a grid scale of 10 km and more than 35 years of observed precipitation, MSWEP v.2.1 provided the best precipitation estimates at the gauge level, the most realistic streamflow simulations (with seven out of ten simulations outperforming the ones obtained using the available precipitation gauge network), and the most realistic simulations of snow cover duration compared to those simulated with the other SPPs.

- SPPs present space-time errors that cannot be assessed when only one indicator and/or time window is used. Indeed, the use of a single indicator is not representative of SPP performance (SPPs may be ranked differently depending on the indicator used) and may conceal part of SPP potential and/or limitation. Similarly, the use of a single time window for SPP assessment may also conceal part of SPP potential and/or limitations (SPPs may rank differently depending on the time window used).

- The proposed sensitivity analysis of snow modelling to the SPPs by using MODIS snow-products as control data has great promise for the assessment of SPP potential over completely unmonitored snow-covered regions.

- For the three assessment indicators (gauges, streamflow and snow cover duration) considered here, all SPP versions including gauge data for precipitation estimates (TMPA-Adj v.7, CMORPH-BLD v.1 and CHIRPS v.2) outperformed their satellite only based version (TMPA-RT v.7, CMORPH-RAW v.1 and CHIRP v.2).

- Soil moisture based precipitation estimates (SM2Rain-CCI v.2) were shown to be very promising precipitation estimates but are unsuitable for regional contexts with large waterbodies, mountainous and snow covered regions, which include too many gaps in time and space. The much smoothed decreasing north-west south-east precipitation

pattern produced by GSMaP v.6 was not sensitive to local precipitation variability, reflecting an overall poor performance.

**Acknowledgements** This work is part of a postdoctoral fellowship funded by the CNES (*Centre National d'Etudes Spatiales*, France). The authors are grateful to SPP and MODIS datasets providers and to the SENAMHI from Bolivia and Peru for providing in situ precipitation, temperature and streamflow observations. The authors are sincerely grateful to the two anonymous reviewers for the time and effort they spent in reading this manuscript and making numerous suggestions which contribute to the study enhancement.

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
