# Peer review of "Consistency of satellite-based precipitation products in space and over time compared with gauge observations and snow-hydrological modelling in the Lake Titicaca region"

_Hydrology and Earth System Sciences, 2018_

## Referee Comment (RC1) · Anonymous Referee #1 · 25 Oct 2018

This is a well-written paper. The analysis is quite thorough although the generalizability of the findings may be limited given the small size of the study region. I recommend publication after a minor revision.

* In Section 1.3, other evaluation studies were criticized for focusing on "a single or a limited sample of SPEs". The current study can be criticized in a similar vein since it ignores reanalysis-based datasets (WFDEI, ERA-Interim, JRA-55, ...). Yes, these datasets have a coarser spatial resolution, but this does not necessarily mean that they perform worse that satellite-based datasets. Yes, I understand that the current

study focuses on satellite-based datasets, but from an end-user perspective the specific source of the precipitation estimates may not matter much.

* The analysis is limited to the 10-day time scale because some of the datasets are only available as daily accumulations starting at midnight UTC. This is a drawback of the study since the daily and subdaily variability is, in my opinion, much more important than the 10-day variability. In addition, focusing only on 10-day means hides much of the difference among the datasets. I understand that the paper is already quite long and I don't expect you to add a daily evaluation but this drawback should at least be mentioned.

* For some of the precipitation datasets it is incorrect and confusing to refer to them as "satellite precipitation estimates (SPEs)". I'm referring to those that are also based on reanalysis and/or gauge data. Consider using "P datasets" instead of "SPEs" throughout the paper.

* Page 11 line 13: "useless" could be misinterpreted. Perhaps "less suitable" or "less well-performing"?

* "Pixel" should be "grid cell" throughout the manuscript.

* Please use perceptually uniform colormaps in the figures. For an explanation, see the following website: https://peterkovesi.com/projects/colourmaps/ . Use a divergent colormap with either white or gray in middle for the %B panels in Figure 5c. This will help the interpretability of the figures.

* The title is pretty long. How about: "Consistency of satellite precipitation estimates with gauge observations and snow hydrological modeling in the Lake Titicaca region"?

* Please use dataset version numbers throughout the manuscript for reproducibility.

* "12 hydrological years (2000-2012)": 12 should be 13 I think.

* The Beck et al. (2017) citation is outdated (the title has changed and the paper is

not in discussion any more). In addition, the citation for MSWEP is wrong (it should be https://journals.ametsoc.org/doi/10.1175/BAMS-D-17-0138.1).

---

## Referee Comment (RC2) · Anonymous Referee #2 · 22 Nov 2018

The manuscript presents an evaluation of several satellite-based precipitation products over a watershed in the Central Andes, using three different evaluation methods: (1) direct comparison with rain gauge data; (2) evaluation of the Nash-Sutcliffe efficiency of the calibration of a hydrological model forced with each precipitation data set on observed stream flow; and (3) evaluation of a hydrological model's capacity to predict snow cover when forced with each of the products. The study finds a quite consistent performance of the products over the three methods, with in general more recent and higher-resolution products performing best.

I agree that there is a need for better insights in the performance of satellite-based precipitation products, especially over complicated and data scarce terrains such as the central Andes. This makes the study relevant from both a scientific and operational perspective.

However, despite the manuscript's length, I do not think that the presented findings are significant enough to merit publication. The main issue is that the evaluation methods applied in the study are not designed to gain any insights in the underlying processes that differentiate the different products. Instead, the evaluations seem to be selected based on two specific applications: predicting streamflow in medium-sized watersheds, and predicting high-elevation snow cover. I can see how those applications may be very relevant in a local operational context, but they add very little to the scientific understanding of satellite precipitation and how satellite-gauge merging algorithms can be improved.

Indeed, despite the author's claim to present a new "protocol" for evaluation, the applied methods are very common (apart from perhaps the snow prediction) and some of the implementation decisions further reduce their capacity to gain process insights, in particular:

- The temporal aggregation to a 10-day period essentially reduces the test to an evaluation of the bias and seasonality of the satellite products, eliminating any insights in their capacity to capture individual events and higher intensities, and their propensity for false alarms.

- Focusing only on the calibration of the hydrological model provides little added value to the direct comparison with rainfall. I agree that a comparison with discharge data is warranted, as the provide a useful independent data set that makes it possible to evaluate potential biases of the precipitation product over a catchment area. This is potentially useful, because of the inherent weakness of comparing point rain gauge data with pixel-average SPEs. However this benefit reduces with increasing size of

the rain gauge network. At the same time a hydrological model-based evaluation has other issues, such as errors in the model structure and ET estimates. None of this is discussed in detail.

- the decision to exclude elevation as a co-variable in the interpolation process nor to analyse the elevation-dependence of SPE performance explicitly, seems a wasted opportunity. Indeed, one of the main advantages of the study region is to understand the performance of the satellite products as a function of topographic characteristics. The cited study that shows that hydrological models are not very sensitive to elevation-dependent rainfall interpolation is in my opinion not a valid argument. In the case of SPEs there are good arguments to expect an elevation-dependent performance.

- I am afraid that I fail to see the purpose of the evaluation using remotely sensed snow cover data. I agree that solid precipitation is a major issue in SPEs that needs to be studied further, and that the study region would be an excellent opportunity to do so. I also agree that SPEs may be used to model river basins with snow cover, where such issues may propagate. But the implementation presented here does not generate any significant insights in either of these issues and I am really left wondering how can be learned from the presented results.

Because if these issues, I think that the current manuscript has only very limited scientific significance beyond the local scale, in the sense that none of the conclusions are sufficiently solid to gain significant insight in how the products may perform in other regions.

At the same time, the paper is very long and contains a lot of information that is readily available in the relevant literature and does not need to be repeated here. In fact, I think that the manuscript could easily be condensed into 1/2 or even 1/3 of the current length. This would make it much sharper and easier to assimilate the presented information.

However, in addition I think that it needs further analysis to increases the scientific significance and provides a more integrated and purposeful evaluation, as opposed to

the current combination of methods, which feel disjoint and ad-hoc. In my opinion this is would have to be a (very) major revision.

Some other issues I identify are:

- I think that it is a missed opportunity not to include IMERG. Of course IMERG does not go as far back in time as the other products. But given that the temporal analysis does not yield much of a trend, I don't think that that is a major issue. At the same time, the added value of the GPM data probably makes it currently one of the most relevant products from a water resources management perspective.

- The language is often rather imprecise, which may lead to misinterpretation. I did not have the time to make an exhaustive list, but a couple of examples include:

p12/8: "overloading": I don't think that you can "overload" research with "unnecessary" results of "useless" SPEs. Al this sounds rather unscientific. It is fair that a subset of product s is used to reduce the workload, but ideally on a scientifically sound and transparent basis.

p12/10: "Pref is influenced by the interpolation process": This is rather euphemistic as Pref is the direct result of the interpolation process.

p13/13: "regional" -> "spatially averaged"

P32/21: "globally": do you mean over the study region? This is surely not global?

Some other specific comments:

p8/6: no orographic effect: I don't follow the reasoning here. Why would accounting for the orographic effect not allow for an objective comparison?

p11/13: "useless SPEs": "useless is quite a strong word. Surely they are not useless for some applications. Perhaps a more scientific formulation can be found?

p12/13: "when more than 80% of daily values records were available": This is not very

conservative and may lead to large errors. I suggest to take only 10-day periods that have complete daily records.

---

## Author Comment (AC1) · 21 Dec 2018

We are sincerely grateful to the two anonymous reviewers for the time and effort spent in reading this manuscript and making numerous suggestions for improvement. The paper has been substantially revised based on their comments. We acknowledge all the points raised and we have spent time on carefully addressing them all. See our answers below. We also wish to inform you about an additional change concerning one of the Satellite-Based Precipitation Estimates (SPPs). We originally considered SM2Rain-CCI v.1, but, after receiving a personal communication from data developers,

we now use the updated version SM2Rain-CCI v.2.

Answer to comments from Referee 1

On "Consistency of satellite precipitation estimates in space and over time compared with gauge observations and snow-hydrological modelling in the Lake Titicaca region" Referee's comment This is a well-written paper. The analysis is quite thorough although the generalizability of the findings may be limited given the small size of the study region. I recommend publication after a minor revision.

Authors' response:

Thank you.

Referee's comment:

* In Section 1.3, other evaluation studies were criticized for focusing on "a single or a limited sample of SPEs". The current study can be criticized in a similar vein since it ignores reanalysis-based datasets (WFDEI, ERA-Interim, JRA-55, ...). Yes, these datasets have a coarser spatial resolution, but this does not necessarily mean that they perform worse that satellite-based datasets. Yes, I understand that the current study focuses on satellite-based datasets, but from an end-user perspective the specific source of the precipitation estimates may not matter much.

Authors' response:

Indeed, the current study focuses on satellite-based datasets and we wanted to test a wide range of products to make a consistent comparison in space and over time. Of course reanalysis-based datasets (WFDEI, ERA-Interim, JRA-55, etc.) could also have been used, since from the point of view of the end user, the actual source of the precipitation estimates may not matter much. However, as you mention, the reanalysis-based datasets have a coarser spatial resolution. Due to the size of our study region, we consequently preferred datasets with spatial resolution equal to $0.25°$ or higher.

Modifications to manuscript:

This is now better explained in the manuscript in the section 2.3.1, page 7, lines 19-22. "It is worth mentioning that other precipitation datasets with coarser resolution (>0.25°) are currently available but we did not use them because (1) the difference between point-gauge and grid-cell-average measurement would introduce inconsistency and because (2) such low resolution datasets are better suited for observation of global scale precipitation patterns rather than the local dynamics studied here."

Referee's comment:

* The analysis is limited to the 10-day time scale because some of the datasets are only available as daily accumulations starting at midnight UTC. This is a drawback of the study since the daily and subdaily variability is, in my opinion, much more important than the 10-day variability. In addition, focusing only on 10-day means hides much of the difference among the datasets. I understand that the paper is already quite long and I don't expect you to add a daily evaluation but this drawback should at least be mentioned.

Authors' response:

The 10-day time scale was used for several reasons. As you saw, some of the datasets we used are only available at a daily time step with a daily accumulation period, which does not match the periods used by the gauges. This difference in timing would be expected to result in differences between the reference and assessed datasets. In addition, because precipitation is subject to high spatial variability, many precipitation events detected at the grid-cell scale would not necessarily be detected at the point-gauge level. This is even more pronounced when only one gauge is used for comparison with the corresponding average pixel measurement (which is mostly the case for the pixels considered here due to the scarcity of local gauges). Another reason is the snow modelling analysis. In this part of the protocol, snow cover distribution (SCD) derived from gap-filled MODIS snow products are used as reference. The temporal filters applied to fill the initial gaps in the MODIS snow products (see section 2.3.2 page 10 line 4-11) led us to consider that these data were more valid at the 10-day than the daily scale, as explained in the manuscript (section 3, page 10, lines 27-33) However, it should be noted that despite the 10-day time scale, significant differences could appear between the satellite-based datasets for each indicator considered in space and over time. This demonstrates that the 10-day time scale enables consistent comparisons.

Modifications to manuscript:

In response to your comment, we have added a few lines to section 3, page 10, lines 25-32 to better explain why we decided to use the 10-day time scale analysis.

"It is noteworthy that the use of a 10-day time scale rather than a daily time scale may conceal some of the differences among the datasets, notably by eliminating any insights into their capacity to capture individual events and higher intensities. However, our choice was based on the inconsistencies we expected between gauges and daily measurements of SPPs as a reason to (i) use a different daily time window aggregation than the local one (8 am to 8 pm) for SPPs delivered at daily scale, (ii) the spatial inconsistency between point-gauge measurement and average grid-cell measurement (Tang et al, 2018), and (iii) the temporal filters used for gap filling of MODIS snow products, which led us to consider that these reference data were more valid at a 10-day scale than at a daily scale."

Referee's comment:

\* For some of the precipitation datasets it is incorrect and confusing to refer to them as "satellite precipitation estimates (SPEs)". I'm referring to those that are also based on reanalysis and/or gauge data. Consider using "P datasets" instead of "SPEs" throughout the paper.

Authors' response:

We agree that the use of "satellite precipitation estimates (SPEs)" could be confusing.

But as all the datasets we used mostly rely on satellite data, we would like to keep the keyword "satellite" in the paper. We thus propose to use "satellite-based precipitation products (SPPs)" to avoid any confusion.

Modifications to manuscript:

The term "Satellite Precipitation Estimates (SPEs) has been replaced by satellite-based precipitation products (SPPs) throughout the manuscript and in the figures.

Referee's comment:

* Page 11 line 13: "useless" could be misinterpreted. Perhaps "less suitable" or "less well-performing"?

Authors' response:

Done. We have changed "useless" to "less suitable".

Referee's comment:

* "Pixel" should be "grid cell" throughout the manuscript.

Authors' response:

Done. We have changed "pixel" to "grid-cell" throughout.

Referee's comment:

* Please use perceptually uniform colormaps in the figures. For an explanation, see the following website: https://peterkovesi.com/projects/colourmaps/. Use a divergent colormap with either white or gray in middle for the %B panels in Figure 5c. This will help the interpretability of the figures.

Authors' response:

Done. We have changed the color in Figure 5 c to grey for "unbiased" estimates (-15% to 15%) which facilitates the observation of systematically precipitation over/under

estimation by the SPPs. We have also changed the CC and CRMSE color scale to green, yellow, orange and red to refer to the four classes considered (<0.5, 0.5-0.6, 0.6-0.7 and >0.7). We also changed the color scale used in figure 6 to green, yellow, orange and red to refer to the four scales considered (<0.5, 0.5-0.6, 0.6-0.7 and >0.7).

Referee's comment:

* The title is pretty long. How about: "Consistency of satellite precipitation estimates with gauge observations and snow hydrological modeling in the Lake Titicaca region"?

Authors' response:

The words "over time and space" are very important as they express part of the originality of the current paper. We would consequently prefer to keep the title in its original form, but replace "satellite precipitation estimates" by "satellite-based precipitation products" based on your preceding comment.

Modifications to manuscript:

The title has been changed to "Consistency of satellite-based precipitation products in space and over time compared with gauge observations and snow-hydrological modelling in the Lake Titicaca region

Referee's comment:

* Please use dataset version numbers throughout the manuscript for reproducibility.

Authors' response:

Done. We now include dataset version numbers throughout the manuscript.

Referee's comment:

* "12 hydrological years (2000-2012)": 12 should be 13 I think.

Authors' response:

In our analysis, we use hydrological years and not calendar years. Therefore, 2000-2012 refers to the period from the 1st of October 2000 to the 30th of September 2012, i.e. 12 hydrological years.

Modifications to manuscript:

Based on your comment, we have added a few lines to clarify it in section 3, page 10 line 21-23: "Each assessment step was analyzed according to three 4-year time windows (2000-2004, 2004-2008, 2008-2012) and one 12-year time window (2000-2012) in which a hydrological year corresponds to a period starting on 1st of October to the following 30th of September."

Referee's comment:

* The Beck et al. (2017) citation is outdated (the title has changed and the paper is not in discussion any more). In addition, the citation for MSWEP is wrong (it should be https://journals.ametsoc.org/doi/10.1175/BAMS-D-17-0138.1)

Authors' response:

We originally cited another article by the same authors which also appears in the discussion. We have changed the reference as follows:

Beck, H. E., Vergopolan, N., Pan, M., Levizzani, V., van Dijk, A. I. J. M., Weedon, G., Brocca, L., Pappenberger, F., Huffman, G. J. and Wood, E. F.: Global-scale evaluation of 23 precipitation datasets using gauge observations and hydrological modeling, Hydrol. Earth Syst. Sci., 6201-6217,https://doi.org/10.5194/hess-21-6201-2017, 2017.

Modifications to manuscript:

We have also added the reference you suggest referring to MSWEP dataset to Table 1.

References :

Tang, G., Behrangi, A., Long, D., Li, C. and Hong, Y.: Accounting for spatiotemporal errors of gauges: A critical step to evaluate gridded precipitation products, J. Hydrol., 559, 294–306, doi:10.1016/j.jhydrol.2018.02.057, 2018.
* * *

---

## Author Comment (AC2) · 21 Dec 2018

We are sincerely grateful to the two anonymous reviewers for the time and effort spent in reading this manuscript and making numerous suggestions for improvement. The paper has been substantially revised based on their comments. We acknowledge all the points raised and we have spent time on carefully addressing them all. See our answers below. We also wish to inform you about an additional change concerning one of the Satellite-Based Precipitation Estimates (SPPs). We originally considered SM2Rain-CCI v.1, but, after receiving a personal communication from data developers,

we now use the updated version SM2Rain-CCI v.2.

Responses to comments from Referee 2

Referee's comment:

The manuscript presents an evaluation of several satellite-based precipitation products over a watershed in the Central Andes, using three different evaluation methods: (1) direct comparison with rain gauge data; (2) evaluation of the Nash-Sutcliffe efficiency of the calibration of a hydrological model forced with each precipitation data set on observed stream flow; and (3) evaluation of a hydrological model's capacity to predict snow cover when forced with each of the products. The study finds a quite consistent performance of the products over the three methods, with in general more recent and higher-resolution products performing best. I agree that there is a need for better insights in the performance of satellite-based precipitation products, especially over complicated and data scarce terrains such as the central Andes. This makes the study relevant from both a scientific and operational perspective.

Authors' response:

Thank you for the time and effort you spent reading the manuscript.

Referee's comment:

However, despite the manuscript's length, I do not think that the presented findings are significant enough to merit publication. The main issue is that the evaluation methods applied in the study are not designed to gain any insights in the underlying processes that differentiate the different products. Instead, the evaluations seem to be selected based on two specific applications: predicting streamflow in medium-sized watersheds, and predicting high-elevation snow cover. I can see how those applications may be very relevant in a local operational context, but they add very little to the scientific understanding of satellite precipitation and how satellite-gauge merging algorithms can be improved.

Authors' response:

As indicated in the title and at the end of the introduction, the aim of this paper is to report on the performance of SPPs (Satellite-based Precipitation Products to replace Satellite Precipitation Estimates used in the previous version) in space and over time compared with gauge observations and snow-hydrological modelling (i.e. from the point of view of the end-user for the management of water resources) rather than to provide insights into the respective techniques and algorithms behind their estimates. As we mention in the introduction, previous studies reporting on the potential of SPPs usually based their methodology on a single indicator (compared with precipitation gauges or the sensitivity of hydrological modelling to SPPs) and on a single time window. However, as underlined in the introduction and in our findings, SPP performance can vary with the indicators, the time window, and the specific region or sub-region analyzed. Substantial adjustments have been made to the abstract, introduction, method and conclusion (see following comment) to clarify the study objectives and main findings. We acknowledge that the added value of our analyses could be limited for readers who are deeply involved in improving SPE algorithms but we believe that these readers will search for that kind of information in journals specialized in remote sensing techniques, which incidentally, have already reported on these topics.

Referee's comment:

Indeed, despite the author's claim to present a new "protocol" for evaluation, the applied methods are very common (apart from perhaps the snow prediction) and some of the implementation decisions further reduce their capacity to gain process insights, . . .

Authors' response:

As mentioned above, the originality of our work is combining different indicators, spatial scales and time windows to test the performance of SPPs. This is clearly a new way to assess SPP consistency (see introduction, section 1.3). The present study investigates the influence of the selected indicators and time windows on the assessment

of the space-time consistency SPPs. The method relies on a comprehensive protocol based on different indicators (gauge observations, streamflow observations via sensitivity analysis of a lumped hydrological model in different catchments, and snow cover observed from satellite imagery via sensitivity analysis of a distributed snow model in an unmonitored mountainous area) applied to four different time windows between 2000 and 2012. Although the individual methods (gauge and hydrological assessment) we use for evaluation are common (apart from the evaluation of the snow cover as you acknowledge), the originality of the protocol is the combination of different indicators, spatial scales and time windows to assess SPPs. As a result of these combinations, we clearly show that the use of a single indicator is not sufficient to assess SPP potential (SPPs rank differently depending on the indicator) and may hide part of SPP potential and/or limitations. What is more, as we demonstrate in this paper, the use of a single time window to assess SPPs may also hide part of the potential and/or limitations depending on the time window considered.

Modifications to manuscript:

Your comment made us aware that the study objectives were not sufficiently clear and we have consequently modified the introduction, part 1.4:

"From the previously established state of the art, this paper investigates the influence of selected indicators and time windows on assessments of the space-time consistency of SPPs. The comprehensive protocol relies on different indicators: gauge observations; (ii) observations of streamflow using sensitivity analysis of a lumped hydrological model in different catchments; and (iii) snow cover observed from satellite imagery via sensitivity analysis of a distributed snow model in an unmonitored mountainous area, applied to four time windows. The aim of using different indicators was to evaluate whether the efficiency of the SPPs varies with the assessment method, whereas different time windows are used to evaluate a potential variation in SPP performance over time. The Lake Titicaca region was selected as study area because it includes all the specific features considered as potential limiting factors for SPPs (high mountain massifs, large water bodies and snow covered areas) to evaluate the potential of SPPs in an extreme context in terms of the sensors' limitations with respect to the orographic effect (i.e. mountains) and high temperature/ emissivity contrast (i.e. Lake Titicaca and a snow-covered region). It also offers the opportunity to provide feedback on the use of SPPs over poorly monitored regions."

We have also modified part of the original abstract from lines page 1 lines 25-32:

". . . whereas the SPP's ability to reproduce the duration of MODIS-based snow cover resulted in poorer simulations than simulation using available precipitation gauges. Interestingly, the potential of the SPPs varied significantly when they are used to reproduce gauge precipitation estimates, streamflow observations or snow cover duration and depending on the time window considered. SPPs thus produce space-time errors that cannot be assessed when a single indicator and/or time windows is used, underlining the importance of carefully considering their space-time consistency before using them for hydro-climatic studies. Among all the SPPs assessed, MSWEP v.2.1 showed the highest space-time accuracy and consistency in reproducing gauge precipitation estimates, streamflow and snow cover duration."

We also made modifications in section 3 page 10 line 21-23:

"Each assessment step was analyzed according to three 4-year time windows (2000-2004, 2004-2008, 2008-2012) and one 12-year time window (2000-2012) in which a hydrological year corresponds to a period starting on 1st of October to the following 30th of September. The aim of the proposed protocol was to investigate the influence of the selected indicator (gauges, streamflow modelling, snow modelling) and time window to assess the SPPs space-time consistency. More details of the proposed protocol are presented in the following sections."

We also made some modifications to the conclusion page 32 line 4-8:

‒ "SPPs present space-time errors that cannot be assessed when only one indicator

and/or time window is used. Indeed, the use of a single indicator is not representative of SPP performance (SPPs may be ranked differently depending on the indicator used) and may conceal part of SPP potential and/or limitation. Similarly, the use of a single time window for SPP assessment may also conceal part of SPP potential and/or limitations (SPPs may rank differently depending on the time window used). Referee's comment

. . . in particular:

1 - The temporal aggregation to a 10-day period essentially reduces the test to an evaluation of the bias and seasonality of the satellite products, eliminating any insights in their capacity to capture individual events and higher intensities, and their propensity for false alarms.

Authors' response :

Referee 1 made a similar comment. We agree that using a 10-day time scale may conceal part of the difference among the datasets, notably by eliminating any insights into their ability to capture individual events and higher intensities. However, we chose the 10-day time scale for several reasons. First, some of the datasets are only available at a daily time step with a daily accumulation period, which does not match the gauge datasets. Inconsistencies between the reference and the assessed datasets are thus to be expected because of this temporal discrepancy. Secondly, as precipitation is characterized by high spatial variability, many precipitation events detected at the grid-cell scale may not be detected at the point-gauge scale (Satgé et al., 2016). This is even more obvious when a single gauge is used for comparison with the corresponding average grid-cell measurement (which is mostly the case for the pixels considered due to the scarcity of local gauges) (see the comprehensive study by Tang et al, 2018). This will inevitably increase the likelihood of false alarms (as suggested) not because of SPP deficiencies but because of the different spatial scale between point-gauges and grid-cell average measurements (see section 5.2 page 28 lines 19-25). Finally,

another reason is the snow modelling analysis. In this part of the protocol, snow cover distribution (SCD) derived from gap-filled MODIS snow products is used as reference. The temporal filters used to fill the initial gaps in the MODIS snow products (see section 2.3.2 page 10 lines 4-11) led us to consider that these data were more valid at the 10-day than at the daily scale. However, it should be noted that evidence for significant differences between the various satellite-based datasets can be found for any space and time indicator despite the 10-day time scale. This shows that the 10-day time scale is consistent in the comparison.

Modifications to manuscript:

Based on your comment, we have added a few lines in section 3 (page 10, lines 25-32) to better explain why we chose the 10-day time scale for analysis.

"It is noteworthy that the use of a 10-day time scale rather than a daily time scale may conceal some of the differences among the datasets, notably by eliminating any insights into their capacity to capture individual events and higher intensities. However, our choice was based on the inconsistencies we expected between gauges and daily measurements of SPPs as a reason to (i) use a different daily time window aggregation than the local one (8 am to 8 pm) for SPPs delivered at daily scale, (ii) the spatial inconsistency between point-gauge measurement and average grid-cell measurement (Tang et al, 2018), and (iii) the temporal filters used for gap filling of MODIS snow products, which led us to consider that these reference data were more valid at a 10-day scale than at a daily scale."

Referee's comment:

2 - Focusing only on the calibration of the hydrological model provides little added value to the direct comparison with rainfall. I agree that a comparison with discharge data is warranted, as the provide a useful independent data set that makes it possible to evaluate potential biases of the precipitation product over a catchment area. This is potentially useful, because of the inherent weakness of comparing point rain gauge

data with pixel-average SPEs. However this benefit reduces with increasing size of the rain gauge network. At the same time a hydrological model-based evaluation has other issues, such as errors in the model structure and ET estimates. None of this is discussed in detail.

Authors' response:

Indeed, as you say, assessments relying on discharge are warranted not only because they are independent datasets, but above all, because discharges incorporate precipitation over a catchment. It is known that precipitation gauge networks frequently do not fully capture spatial precipitation features, even more frequently in regions where data are scarce. The purpose in our study is to highlight the complementarity of point gauge comparison and the sensitivity analysis of hydrological modelling (see section 5.2 "Gauges versus hydrological modelling-based assessment", page 27 line 20 to page 28 line 7). We used model calibration to report on hydrological model sensitivity to different SPP forcing data. Independent validation of the hydrological model is beyond the scope of this paper, as our aim was not to assess the ability of the model to reproduce streamflow responses in detail nor to be transferred under climate variability in the basins studied. Similarly, our purpose is not to evaluate and discuss model structure or ET estimates. Consequently, the same ET estimates and hydrological model were used to assess all the SPPs considered by using a specific calibration for each dataset.

Modifications to manuscript:

We believe your comment was due to the fact our objectives were not clear enough. As mentioned above, they have been clarified in the revised version. We have also added further details in the discussion (see section 5.2 page 29 lines 8-12).

"Similarly, other ET estimates could have been used, especially those based on remote sensing techniques to compensate for the limited availability of temperature gauge data. However, we would like to underline that our purpose was not to evaluate or

discuss the model structure or ET estimates, but to highlight the complementarity of point gauges and integrated hydrological modelling. In this context, the same ET estimates and the same hydrological model were used to assess all the SPEs through the specific calibration of each dataset."

Referee's comment:

3 - The decision to exclude elevation as a co-variable in the interpolation process nor to analyse the elevation-dependence of SPE performance explicitly, seems a wasted opportunity. Indeed, one of the main advantages of the study region is to understand the performance of the satellite products as a function of topographic characteristics. The cited study that shows that hydrological models are not very sensitive to elevation dependent rainfall interpolation is in my opinion not a valid argument. In the case of SPEs there are good arguments to expect an elevation-dependent performance.

Authors' response:

As explained in section 2.2.2, pages 7, lines 13-15, we decided to not take the potential elevation-dependency into account in the interpolation of in-situ precipitation data (unlike temperature). This was justified by the fact that the available gauges are all located on the flat part of the four study basins, which made it impossible to show a potential relationship between elevation and precipitation in these basins. However, it should be noted that the actual elevation-dependency potential accounted for in the SPE was implicitly evaluated through hydrological modelling of this topographically complex region.

It is also worth mentioning that, as reported in many studies, SPPs are generally less accurate over mountainous regions (see discussion on this point in section 5.4, page 30, lines 20-27). Two publications referring to the same region already reported on SPP accuracy as a function of elevation (Ochoa et al., 2014; Satgé et al., 2017b).

Modifications to manuscript:

[Figure]

The decision to exclude elevation as a co-variable in the interpolation of in-situ precipitation has been reformulated in section 2.2.2, pages 7, lines 13-15, to clarify our purpose with respect to this point. "Because the gauges are mainly located in the flat land part of the basins, it was not possible to provide evidence for an effect of elevation on precipitation distribution. Consequently, no orographic effect was accounted for in the interpolation of the point precipitation observations"

Referee's comment:

4 - I am afraid that I fail to see the purpose of the evaluation using remotely sensed snow cover data. I agree that solid precipitation is a major issue in SPEs that needs to be studied further, and that the study region would be an excellent opportunity to do so. I also agree that SPEs may be used to model river basins with snow cover, where such issues may propagate. But the implementation presented here does not generate any significant insights in either of these issues and I am really left wondering how can be learned from the presented results.

Authors' response:

Considering snow-cover as a reference to evaluate SPP consistency is a promising new way to differentiate rainfall and snowfall over unmonitored regions. It is part of the effort to cross-validate remote sensing datasets. In this context, a recent study reports on the possibility of using remote sensing soil humidity datasets to assess SPEs potential (Massari, Crow and Brocca 2017). In the case of the Altiplano, we found no evidence of significant differences in performance between the different precipitation datasets (Pref and SPPs) using the snow modelling approach. This was a bit disappointing and is further discussed in section 5.4. (page 31 line 1-5). The absence of clearer differences may be explained by the presence of permanent snow-covered areas whose seasonal dynamics are hard to capture from optical imagery (MODIS), because there is more variation in depth than in areal extent. The regional seasonal snow cycle captured by MODIS is consequently weak and erratic. However, the results

reported here do not invalidate the proposed protocol. In other regions with other snow conditions, it is a promising way to assess SPPs thanks to the sensitivity of snow modelling in reproducing snow cover dynamics as observed from MODIS snow-products. We believe that many readers will find the proposed method useful for their studies.

Referee's comment:

Because if these issues, I think that the current manuscript has only very limited scientific significance beyond the local scale, in the sense that none of the conclusions are sufficiently solid to gain significant insight in how the products may perform in other regions.

Authors' response:

Indeed, as stated in the introduction, the consistency of SPPs varies with the region because it depends on (i) land cover (emissivity, surface temperature), (2) topography, and (3) precipitation intensities. It is therefore correct to say that the conclusions on the SPPs drawn from this study cannot be extrapolated to other regions of the world. However, as mentioned above, the main scientific interest of the study is the protocol we propose to assess SPP consistency. Indeed, the aim of this protocol is to show how the current assessment of SPP potential can be influenced by the method used (single indicator, single time window), and, as a consequence, to recommend the use of different indicators to provide consistent reports on SPPs potential/limits. Finally, by identifying and elucidating these issues, the paper also reports on the extent to which SPEs can be used in poorly monitored basins to support water resources management.

Modifications to manuscript:

As a result of this comment (and of an earlier comment you made), we realized that the objectives of our study were not clear and we have consequently considerably modified the introduction part 1.4 as follows:

"From the previously established state of the art, this paper investigates the influence of

selected indicators and time windows on assessments of the space-time consistency of SPPs. The comprehensive protocol relies on different indicators: gauge observations; (ii) observations of streamflow using sensitivity analysis of a lumped hydrological model in different catchments; and (iii) snow cover observed from satellite imagery via sensitivity analysis of a distributed snow model in an unmonitored mountainous area, applied to four time windows. The aim of using different indicators was to evaluate whether the efficiency of the SPPs varies with the assessment method, whereas different time windows are used to evaluate a potential variation in SPP performance over time. The Lake Titicaca region was selected as study area because it includes all the specific features considered as potential limiting factors for SPPs (high mountain massifs, large water bodies and snow covered areas) to evaluate the potential of SPPs in an extreme context in terms of the sensors' limitations with respect to the orographic effect (i.e. mountains) and high temperature/ emissivity contrast (i.e. Lake Titicaca and a snow-covered region). It also offers the opportunity to provide feedback on the use of SPPs over poorly monitored regions."

We have also modified part of the original abstract from lines page 1 lines 25-32:

". . . whereas the SPP's ability to reproduce the duration of MODIS-based snow cover resulted in poorer simulations than simulation using available precipitation gauges. Interestingly, the potential of the SPPs varied significantly when they are used to reproduce gauge precipitation estimates, streamflow observations or snow cover duration and depending on the time window considered. SPPs thus produce space-time errors that cannot be assessed when a single indicator and/or time windows is used, underlining the importance of carefully considering their space-time consistency before using them for hydro-climatic studies. Among all the SPPs assessed, MSWEP v.2.1 showed the highest space-time accuracy and consistency in reproducing gauge precipitation estimates, streamflow and snow cover duration."

We also made modifications in section 3 page 10 line 21-23:

"Each assessment step was analyzed according to three 4-year time windows (2000-2004, 2004-2008, 2008‒2012) and one 12-year time window (2000-2012) in which a hydrological year corresponds to a period starting on 1st of October to the following 30th of September. The aim of the proposed protocol was to investigate the influence of the selected indicator (gauges, streamflow modelling, snow modelling) and time window to assess the SPPs space-time consistency. More details of the proposed protocol are presented in the following sections."

We also made some modifications to the conclusion page 32 line 4-8:

"SPPs present space-time errors that cannot be assessed when only one indicator and/or time window is used. Indeed, the use of a single indicator is not representative of SPP performance (SPPs may be ranked differently depending on the indicator used) and may conceal part of SPP potential and/or limitation. Similarly, the use of a single time window for SPP assessment may also conceal part of SPP potential and/or limitations (SPPs may rank differently depending on the time window used).

Referee's comment:

At the same time, the paper is very long and contains a lot of information that is readily available in the relevant literature and does not need to be repeated here. In fact, I think that the manuscript could easily be condensed into 1/2 or even 1/3 of the current length. This would make it much sharper and easier to assimilate the presented information.

Authors' response:

We can understand why you feel the paper is very long. However we address readers in both the hydrology and remote sensing communities, so shortening it by half or even by one third does not seem reasonable. We believe it is important to provide sufficient detail so it can be understood all.

Modifications to manuscript:

Without more detailed information on what you think should be shortened, we have

limited our reductions to a few sentences or parts of sentences. An exhaustive list is given below:

Page 1, line 23, we deleted: "by the hydrological model tested"

Page 2, lines 4-5, we deleted: "After some adjustment over Lake Titicaca, MSWEP v.2.1 should thus be preferred for the regional hydro-meteorological survey."

Page 2, line 29, we deleted: "as precipitation estimates were found to be very accurate"

Section 2.1 Study Area. The following paragraph has been deleted:

"Runoff from the lake basin and direct precipitation over the lake contribute approximately 53 and 47% of the lake water supply, respectively (Roche et al., 1992). However, studies on the regional water balance are more than 30 years old (Carmouze et al., 1977; Carmouze and Aquize, 1981; Lozada, 1985) and may not be representative of the current situation. Indeed, a recent study over the Altiplano (López-Moreno et al., 2015) found a temperature increase of between 0.15 and 0.25°C decade-1 between 1965 and 2012, while Heidinger et al. (2018) reported an increase in the intensity of precipitation extremes over the period 1965-2010 in the same region. The temperature is expected to continue to increase until the end of this century while precipitation may decrease by 10 to 30% depending on the climate projections (Bradley, 2006; Minvielle and Garreaud, 2011; Urrutia and Vuille, 2009). There is thus a need to update the Lake Titicaca water balance to support efficient water management to adapt to this changing situation. However, the transboundary, economic and remote context means the sparse hydro-meteorological monitoring network prevents a consistent analysis. Indeed, precipitation presents high spatiotemporal variability due to the geomorphologic and tropical context which is poorly represented by the available monitoring network. With nearly-global scale coverage, SPPs are a promising alternative to monitor regional precipitation in space and over time."

And replaced it by:

"The Lake Titicaca is drained by the Desaguadero River to the south (Fig. 1) which contributes up to 65% of water inflows into Lake Poopó (second largest Bolivian lake) (Pillco and Bengtsson, 2010). An accurate Lake Titicaca water balance for monitoring purposes is therefore crucial to support efficient water resources management in the Altiplano. However, the transboundary, economic and remote context means hydro-meteorological monitoring is sparse. Thanks to almost global scale coverage, SPPs represent a promising alternative to monitor regional precipitation in space and over time, and offer unprecedented opportunity to achieve efficient regional water resources management."

Page 8 line 4, we deleted: "for precipitation estimates"

Page 8 line 11, we deleted: "This is why we aimed to analyse SPPs at a 10-day time step for which we assumed that differences in daily time windows were negligible."

Page 11 line 11, we deleted "To avoid overloading the analysis with less suitable SPPs"

Page 17 line 5, we deleted: "Figure 5a is the Taylor diagram obtained from SPPs at the regional scale for the 2000-2012 period."

Page 18 line 3, we deleted: "Figure 5b presents the performance of the SPPs at the regional scale for the three 4-year time windows considered: 2000-2004, 2004-2008 and 2008-2012, in the form of a Taylor diagram."

Page 19 line 6, we deleted: "Figure 6 shows the performance of the SPPs for three 4-year periods (2000-2004, 2004-2008, and 2008-2012) in terms of CRMSE."

Page 21 line 6, we deleted: "Figure 7 shows the efficiency (in terms of NSE scores) of the hydrological model in reproducing streamflow at the outlet of the four tested basins (Ilave, Katari, Keka, and Ramis) using Pref and SPPs as input data over the period 2008-2012"

Page 24, line 1, we deleted: "Figure 8 compares the snow cover duration (SCD) ob-served by MODIS and the durations simulated by the snow model using Pref or all the

SPPs as input data over the different periods."

Referee's comment:

However, in addition I think that it needs further analysis to increases the scientific significance and provides a more integrated and purposeful evaluation, as opposed to the current combination of methods, which feel disjoint and ad-hoc. In my opinion this is would have to be a (very) major revision.

Authors' response:

As explained above, our objectives were not clear. Regarding the protocol, each step of it provides complementary information on SPP space-time consistency, which is crucial information for their use and enhancement. We hope that the changes made in the manuscript and the additional information reported will clarify the objectives and relevance of the present study.

Referee's comment:

- I think that it is a missed opportunity not to include IMERG. Of course IMERG does not go as far back in time as the other products. But given that the temporal analysis does not yield much of a trend, I don't think that that is a major issue. At the same time, the added value of the GPM data probably makes it currently one of the most relevant products from a water resources management perspective.

Authors' response:

In fact, we already evaluated GPM based SPPs (IMERG and GSMAP) in a previous study (see Satgé et al, 2017). IMERG only covers the period from 2014 to the present. As assessment based on different time windows is one of the main points of the proposed protocol, it would be impossible to include IMERG. We would also like to point out that the temporal analysis yields much more than a trend. Actually, when looking at potential of SPPs as forcing data in hydrological model, the ranking of SPE performance varied considerably depending on the time window considered (Figure 7). To

give a rapid example, we cite part of the text in section 4.2 "Space-time consistency of SPEs compared with streamflow simulations" (see page 22 lines 8-15):

"However, as shown in Fig. 7c, the SPE hydrological ranking in the 2008-2012 period changed drastically over time. For example, for the Katari catchment, MSWEP v.2.1 led to the best streamflow simulations for the 2004-2008 and 2000-2012 but not for the 2000-2004 period, for which TMPA-Adj v.7 forced streamflow simulations had a higher NSE score of 0.85. Additionally, CMORPH-BLD v.1 potential fell drastically over the 2000-2004 period with a negative NSE score, whereas it produced the most realistic streamflow simulation for the period 2008-2012. In the Keka catchment, for each time window, the best streamflow simulation was obtained using different SPPs. CHIRP v.2, PERSIANN-CDR and CMORPH-BLD v.1 resulted in the highest NSE scores over the various sub-periods analysed, with respectively 0.73 for the period 2000-2004, 0.83 for 2004-2008 and 0.77 for 2008-2012"

Modifications to manuscript:

We realized that this information needed to be recalled in the discussion part 5.3 "Space-time SPP consistency" and we have therefore added a few lines about this in the discussion part 5.3 page 29 line 23 to page 30 line 1.

"For each assessment indicator considered (i.e. gauges, streamflow and snow cover duration), the performances of the SPPs varied depending on the time window considered. These variations can be easily observed when SPPs are used as forcing data for hydrological modelling, in which case the ranking of SPPs observed for each of the 2000-2004, 2004-2008, 2008-2012 and 2000-2012 time windows changed significantly (see Fig. 12b and 12c)."

Referee's comment:

- The language is often rather imprecise, which may lead to misinterpretation. I did not have the time to make an exhaustive list, but a couple of examples include:

Authors' response:

We agree. We have responded to each of your comments concerning the language and the text has been corrected by a native speaker.

Referee's comment:

p12/8: "overloading": I don't think that you can "overload" research with "unnecessary" results of "useless" SPEs. Al this sounds rather unscientific. It is fair that a subset of products is used to reduce the workload, but ideally on a scientifically sound and transparent basis.

Authors' response:

Agreed, we changed the text: "To avoid overloading the analysis,.." — > "To filter less suitable SPEs,.." "Useless" — > "less suitable"

Referee's comment:

p12/10: "Pref is influenced by the interpolation process": This is rather euphemistic as Pref is the direct result of the interpolation process.

Authors' response:

Agreed. This sentence has been removed.

Referee's comment:

p13/13: "regional" -> "spatially averaged"

Authors' response:

Done. This term has been corrected throughout the text.

Referee's comment:

P32/21: "globally": do you mean over the study region? This is surely not global? Some other specific comments:

Authors' response:

Here we mean "in general". We deleted the word "globally".

Referee's comment:

p8/6: no orographic effect: I don't follow the reasoning here. Why would accounting for the orographic effect not allow for an objective comparison?

Authors' response:

Agreed. See above our answer to your comment #3 on the decision to exclude elevation as a co-variable in the interpolation process.

Modifications to manuscript:

Based on your previous comment, we have changed the text concerning the use of elevation as a co-variable in the precipitation interpolation process (see section 2.2.2, page 7 line 13-15). "Because the gauges are mainly located in the flat land part of the basins, it was not possible to provide evidence for an effect of elevation on precipitation distribution. Consequently, no orographic effect was accounted for in the interpolation of the point precipitation observations".

Referee's comment:

p11/13: "useless SPEs": "useless is quite a strong word. Surely, they are not useless for some applications. Perhaps a more scientific formulation can be found?

Authors' response:

Done. We have replaced 'useless' by "less suitable"

Referee's comment:

p12/13: "when more than 80% of daily values records were available": This is not very conservative and may lead to large errors. I suggest to take only 10-day periods that have complete daily records.

[Figure]

Authors' response:

We agree that this statement was poorly explained. Actually, only 10-day amounts with more than 80% of common daily data available were used and each 10-day amount was computed using only daily amounts available for Pref and all SPPs in order to compare exactly the same thing.

Modifications to manuscript:

In response to your comment, we have changed the text to make it clearer (see section 3.1 page 11 lines 11-14). "For each of the 69 0.05° grid-cells, the 10-day records were only computed when more than 80% of daily values were available from all precipitation datasets (Pref and SPPs) for exactly the same date. Next, mean spatially averaged 10-day precipitation series were computed from Pref and all SPPs by aggregating the values from all 69 grid-cells."

References:

Abiodun, O. O., Guan, H., Post, V. E. A., and Batelaan, O.: Comparison of MODIS and SWAT evapotranspiration over a complex terrain at different spatial scales, Hydrol. Earth Syst. Sci., 22, 2775-2794, https://doi.org/10.5194/hess-22-2775-2018, 2018

Beck, H. E., van Dijk, A. I. J. M., Levizzani, V., Schellekens, J., Miralles, D. G., Martens, B., and de Roo, A.: MSWEP: 3-hourly 0.25° global gridded precipitation (1979–2015) by merging gauge, satellite, and reanalysis data, Hydrol. Earth Syst. Sci., 21, 589-615, https://doi.org/10.5194/hess-21-589-2017, 2017

López, O., Houborg, R., and McCabe, M. F.: Evaluating the hydrological consistency of evaporation products using satellite-based gravity and rainfall data, Hydrol. Earth Syst. Sci., 21, 323-343, https://doi.org/10.5194/hess-21-323-2017, 2017.

Massari, C., Crow, W., and Brocca, L.: An assessment of the performance of global rainfall estimates without ground-based observations, Hydrol. Earth Syst. Sci., 21, 4347-4361, https://doi.org/10.5194/hess-21-4347-2017, 2017

Satgé, F., Bonnet, M.-P., Gosset, M., Molina, J., Hernan Yuque Lima, W., Pillco Zolá, R., Timouk, F. and Garnier, J.: Assessment of satellite rainfall products over the Andean plateau, Atmos. Res., 167, 1–14, doi:10.1016/j.atmosres.2015.07.012, 2016.

Tang, G., Behrangi, A., Long, D., Li, C. and Hong, Y.: Accounting for spatiotemporal errors of gauges: A critical step to evaluate gridded precipitation products, J. Hydrol., 559, 294–306, doi:10.1016/j.jhydrol.2018.02.057, 2018.
* * *

---

## Author Comment (AC3) · 21 Dec 2018

Please find the revised manuscript version in supplement data with the modifications highlighted in yellow.

Please also note the supplement to this comment: https://www.hydrol-earth-syst-sci-discuss.net/hess-2018-316/hess-2018-316-AC3-supplement.pdf

---

## Author Response (AR2)

**Answer to comments from Referee 1 (round 2)**

On "Consistency of satellite precipitation estimates in space and over time compared with gauge observations and snow-hydrological modelling in the Lake Titicaca region"

**General reply**

We are sincerely grateful to the anonymous reviewer. The paper has been substantially revised based on its comments. We acknowledge all the points raised and we have spent time on carefully addressing them all. See our answers below. Changes in the text are highlighted in yellow.

**Referee comment**

*Thank you for making the changes which I believe have resulted in a better paper.*

**Author's response**

Thank you.

**Referee comment**

*I have a few more comments.*

*"It is worth mentioning that other precipitation datasets with coarser resolution (>0.25°) are currently available but we did not use them because (1) the difference between point-gauge and grid-cell-average measurement would introduce inconsistency and because (2) such low resolution datasets are better suited for observation of global scale precipitation patterns rather than the local dynamics studied here."*

*I disagree with this (newly added) statement. There are numerous situations where a coarse resolution dataset (e.g., JRA-55 with its 0.7° resolution) would provide a much better performance than a high resolution dataset (e.g., IMERG with its 0.1° resolution). Why would it matter to an end-user whether there's a potential discrepancy between point-gauge and grid-cell-average measurements? For most applications all that matters is the accuracy of the dataset.*

**Author's response**

We agree that coarse resolution datasets may provide a better performance than high resolution datasets. However, the part of the text you mentioned aim at explain why we did not consider these datasets in our study. We did not use those coarse resolution datasets based on: (1) the available reference gauge network and (2) the specifics of the studied basins.

Firstly, most of the considered grid-cell used to assess SPP potential include only one gauge. As discussed (see discussion 5.2 Page 27 Line 31 to Page28 line 8), a point gauge measurement is not totally representative of a surface area measurement. Therefore, assessing precipitation dataset consistency by comparing reference point measurement with superficial area measurement suffer some inconsistency. This inconsistency will inevitably increase for precipitation dataset with grid size

superior to 0.25° (>625 km$^2$) and especially for JRA-55 and its 0.7° (4900 km$^2$) grid-cell size. Therefore, we believe that assessing coarse resolution dataset with the available gauge network will not be representative of their actual potential.

Secondly, those coarse resolution datasets may not be suited for two of the considered basins. Actually, 0.7° grid size products like JRA-55 present a grid size (4900km$^2$) bigger than the surface area of the Katari and Keka basins (2588 km$^2$ and 801 km$^2$, respectively). Therefore, for these basins the precipitation will be hardly measured from SPPs with grid-cell size superior to their superficial extent. Furthermore, the snow cover dynamic analyses was applied over a superficial extent smaller than the non-considered SPPs grid-cell size. Therefore, coarse resolution SPP will be unsuited to follow the snow cover dynamic extent dynamic over this region. Therefore, the coarse resolution SPPs are not suited in an end users perspective to follow the considered streamflow and/or snow cover dynamics.

**Modifications to the text**

We understand that our choice might be unclear after reading these highlighted lines. We modified the text to:

*"Other precipitation datasets with coarser resolution (>0.25°) are currently available but we did not use them because: (1) the scarce available gauges network will not warrant a consistent potential assessment in reason to the difference between point-gauge and grid-cell-average measurement (Tang et al., 2017) and (2) the considered catchments and snow analyses zone area are smaller than the grid-cell size of such coarse resolution precipitation datasets. However, it is worth mentioning that in specific situations, coarse resolution SPPs could perform better than higher resolution SPPs (Beck et al., 2018) and that reanalyzes precipitation datasets tend to be a better choice in cold regions/periods (Huffman et al., 1995). Such statement cannot be verified in the present study in reason to the scarce gauge network context and considered catchments and snow analysis zone area."*

**Referee comment**

*You should explicitly mention (at least in the introduction and conclusion sections) that reanalyzes tend to be a better choice in cold regions/periods. The latter has been known for several decades (see, e.g., Huffman et al., 1995) but is generally not communicated well in most precipitation dataset evaluation studies which can lead to confusion.*

**Author's response**

We now mention the problem you presented regarding the reanalyzes precipitation datasets and cold regions/period. However, we limit this statement to the methods´ section after justifying why they were not considered but could be relevant in particular cases (see previous comment).

**Referee comment**

*Regarding the figures, my suggestion to use perceptually uniform color scales was not followed. Let me give you another reference: https://arxiv.org/abs/1509.03700. In addition, the use of only red and green in color scales is not recommended as many males have red-green color blindness*

*(https://nei.nih.gov/health/color_blindness/facts_about). The resolution of some of the figures is also a bit low (in Figure 7, for example, the lines are very pixelated).*

**Author's response**

We did not clearly understand what you were expecting for and we are sorry that our changes did not match your request.

We now use a divergent color map for figure 5c (bias) and return to the initial blue color for figure 5c (CC and RMSE) to avoid any problem for blindness readers.

Figure 6 also use now blue color to avoid any problem for blindness readers.

Resolution of Figure 7 was improved from 300 dpi to 450 dpi.

We changed Figure 9 color bar for a "divergent" one.

Note that we did not proceed to any changes in Figure 2 because the current color bar is very effective to represent the regional precipitation pattern.

**Referee comment**

*Finally, "Global-scale evaluation of 23 precipitation datasets" should be "Global-scale evaluation of 22 precipitation datasets", and the Beck et al. (2018) reference is missing from the references list.*

**Author's response**

Actually, in table 1 we miswrote Beck et al., 2017 for Beck et al., 2018. However, we also added the reference Beck et al., 2018 to justify on the potential benefits of coarse resolution dataset (see Page 7 line 22-23):

 *"However, it is worth mentioning that in specific situations, coarse resolution SPPs could perform better than higher resolution SPPs (Beck et al. 2018)"*